# Sources of Surface $O_3$ in the UK: Tagging $O_3$ within WRF-Chem

Johana Romero-Alvarez[1,a,b,c], Aurelia Lupaşcu[2], Douglas Lowe[3,d], Alba Badia[1,e], Scott Archer-Nicholls[4,d], Steve Dorling[1], Claire E. Reeves[1], Tim Butler[2,5]

[1] School of Environmental Sciences, University of East Anglia, Norwich, UK

[2] Institute for Advanced Sustainability Studies (IASS), Potsdam, Germany

[3] Centre for Atmospheric Sciences, School of Earth, Atmospheric and Environmental Sciences, University of Manchester, Manchester, UK

[4] Centre for Atmospheric Science, Department of Chemistry, University of Cambridge, Cambridge, UK

[5] Freie Universität Berlin, Institut für Meteorologie, Berlin, Germany

[a] Now at: Department of Chemistry, University of Colorado Boulder, Boulder, USA

[b] Now at: Cooperative Institute for Research in Environmental Sciences (CIRES), University of Colorado Boulder, Boulder, CO, USA

[c] Now at: NOAA Global Systems Laboratory (GSL), Boulder, CO, USA

[d] Now at Research IT, University of Manchester, Manchester, UK

[e] Now at Institute of Environmental Science and Technology (ICTA), Universitat Autònoma de Barcelona, Barcelona, Spain

*Correspondence to*: Johana Romero Alvarez (lero1992@colorado.edu)

**Abstract.** Tropospheric ozone ($O_3$) concentrations depend on a combination of hemispheric, regional and local-scale processes. Estimates of how much $O_3$ is produced locally vs. transported from further afield are essential in air quality management and regulatory policies. Here, a tagged-ozone mechanism within the WRF-Chem model is used to quantify the contributions to surface $O_3$ in the UK from anthropogenic nitrogen oxide (NOx) emissions from inside and outside the UK during May-August 2015. The contribution of the different source regions to three regulatory $O_3$ metrics is also examined. It is shown that model simulations predict the concentration and spatial distribution of surface $O_3$ with a domain-wide mean bias of -3.7 ppbv. Anthropogenic NOx emissions from the UK and Europe account respectively for 13% and 16% of the monthly mean surface $O_3$ in the UK, as the majority (71%) of $O_3$ originates from the hemispheric background. Hemispheric $O_3$ contributes the most to concentrations in the north and the west of the UK with peaks in May, whereas European and UK contributions are most significant in the east, south-east, and London, i.e., UK's most populated areas, intensifying towards June and July. Moreover, $O_3$ from European sources is generally transported to the UK rather than produced in-situ. It is demonstrated that more stringent emission controls over continental Europe, particularly in western Europe, would be necessary to improve the health-related metric MDA8 $O_3$ above 50 and 60 ppbv. Emission controls over larger areas, such as the northern hemisphere, are instead required to lessen the impacts on ecosystems as quantified by the AOT40 metric.

# 1    Introduction

Tropospheric ozone ($O_3$) is a pollutant of concern for policy-makers because of its detrimental effects on human health, agriculture and ecosystems (Fuhrer, 2009; WHO, 2016). Near ground level, $O_3$ has a typical atmospheric lifetime of a few hours. In the free troposphere, however, the lifetime can be up to several weeks (Stevenson et al., 2006) and $O_3$ can be transported from its point of production downwind over long distances crossing countries and continents (Wild et al., 2004; HTAP, 2007). The concentration of $O_3$ at a given location is therefore dictated by a combination of hemispheric, regional and local-scale factors (Jenkin, 2008). Examples of this are long-range transport of $O_3$ and its precursors, including stratospheric intrusions, and photochemical reactions happening on a local and regional scale (e.g., Monks, 2000; HTAP, 2007).

The production of $O_3$ in the troposphere is highly non-linear. It depends on the abundance of nitrogen oxides (NOx = $NO_2$ + NO) and peroxy radicals ($HO_2$) produced after the oxidation of volatile organic compounds (VOCs) by hydroxyl radical (OH) (Monks, 2005). The reaction of NO with $HO_2$ and the subsequent photolysis of $NO_2$ generating $O_3$ is the primary known mechanism of $O_3$ production (Atkinson, 2000; Monks, 2005). NOx concentrations determine whether $O_3$ is produced or chemically removed (Monks, 2005). In the rural areas of most industrialized countries, where NOx is available at moderate levels, the rate of $O_3$ formation increases with increasing NOx concentrations (NOx-limited regime). In more polluted areas, by contrast, high NOx concentrations inhibit $O_3$ formation as this begins being depleted by NO (NOx titration effect). Subsequent formation of nitric acid ($HNO_3$) from the reaction of $NO_2$ with OH constitutes a major endpoint for $O_3$ in such environments (Monks, 2005). However, elevated inputs of non-methane VOCs (NMVOCs) can increase the production of $O_3$ as the reaction of VOCs with OH radicals become more significant (NOx-saturated regime).

Furthermore, $O_3$ concentrations also depend on its deposition, uptake by vegetation and meteorological variables such as temperature, winds (direction and speed), solar radiation intensity and precipitation (e.g., Sillman, 1999; Coyle et al., 2002). For instance, high $O_3$ concentration episodes in the UK have been associated with heatwave periods (Finch and Palmer, 2020). The contribution of each process varies with location. Remote sites are largely controlled by hemispheric background $O_3$ (AQEG, 2009). Photochemical pollution episodes, on the other hand, are more severe in the south and east of the UK and $O_3$ titration by NOx is higher in urban areas (Jenkin, 2008).

In the UK, tighter UK and European precursor emissions controls in the last 30 years have led to a substantial decrease in the concentration of $O_3$ primary precursors and successfully reduced the severity of the high $O_3$ concentration episodes (AQEG, 2009; Derwent et al., 2018; Finch and Palmer, 2020). Even so, exposure to surface $O_3$ continues to cause considerable damage to human health in Europe and the UK leading to an estimated 17,000 premature deaths in 2015 (EEA, 2017). Evidence suggests that annual mean $O_3$ concentrations in the UK have been increasing in urban/suburban areas and to a lesser extent in rural areas (Jenkin, 2008; AQEG, 2009; Munir et al., 2013; Finch and Palmer, 2020). Reductions in NOx emissions, mainly by road traffic, have led to reductions in the $O_3$ scavenging in urban areas so that $O_3$ concentrations have generally increased (Finch and Palmer, 2020). The increase in rural areas, on the other hand, has been largely driven by rising hemispheric $O_3$ levels, up to +0.31 ppbv a year over the 20-year 1987–2007 period (Derwent et al., 2007) and +0.25 ppb a year over the 25-

year period (Derwent et al., 2013). Accordingly, increasing emissions of precursors in Asia and North America influence $O_3$ concentrations entering Europe from the North Atlantic, offsetting the effects of European regional emission reductions on $O_3$ (HTAP, 2010; Derwent et al., 2018). Therefore, efficient emissions control policies aimed at reducing $O_3$ concentration in a given region require a holistic assessment of both $O_3$ transport from outside the region and in-situ $O_3$ production. Such

quantitative estimations can be made by applying source-receptor methods (S-R) within Chemical Transport Models (CTMs). S-R studies often compare model simulations that include all anthropogenic emissions with those obtained after modifying emissions from a region of interest (the so-called perturbation approach). However, as $O_3$ chemistry is highly non-linear, this approach can lead to unrealistic attribution estimates, e.g., Emmons et al. (2012) underestimated the $O_3$ contribution by up to a factor of 4 when perturbing NO emissions by 20%. Tagged-ozone methods, on the other hand, use additional diagnostics to

follow the reaction of different emissions to the formation of $O_3$, making the approach suited to investigate the contribution of different precursors (Emmons et al., 2012; Grewe et al., 2012; Butler et al., 2018).

Several studies have investigated the contribution of intercontinental transport to $O_3$ in Europe, in particular from North America and Asia, using different modelling techniques (e.g., Li, 2002; Derwent et al., 2004; Auvray and Bey, 2005; Sudo and Akimoto, 2007; Derwent et al., 2008; Emmons et al., 2012; Derwent et al., 2015; Mertens, 2017; Butler et al., 2018;

Lupaşcu and Butler, 2019; Butler et al., 2020). However, these studies do not provide a quantitative estimate of the contribution of the different source regions to the total amount of $O_3$ over the UK at a regional scale, but rather an estimate at a national scale or at individual locations across Europe or for the European region as a whole.

The present study quantifies the contributions to surface $O_3$ in 12 receptor regions in the UK from anthropogenic NOx emissions from inside and outside the UK using the tagged-ozone method developed in Lupaşcu and Butler, (2019). Dividing

the UK into several regions serves to separate meteorological features and chemical environments that are known to impact the spatial distribution and temporal variation of air pollutants such as $O_3$ (Coyle et al., 2002; Jenkin, 2008). We focus on summer 2015 which saw several heat waves causing elevated $O_3$ values in Central and Western Europe that exceeded the EU information threshold of 1 hour (h) average mixing ratio of 90 ppbv (Tarrasoet al., 2016). We also look at the impact of $O_3$ on human exposure, crops, and vegetation using two well-known $O_3$ metrics, the MDA8 and the AOT40. The MDA8 is a health-

related metric commonly used to assess the impacts of $O_3$ exposure on the population (e.g., Reidmiller et al., 2009; Stock et al., 2013; Mar et al., 2016). The metric is defined as the maximum daily 8-hourly (h) average (MDA8) $O_3$ values (in ppbv) and is strongly influenced by photochemical episodes (AQEG, 2009). The AOT40 (accumulated $O_3$ above a threshold of 40 ppbv) is commonly used to assess the effects of $O_3$ on crops and vegetation and is based on exposure over 40 ppbv using only the 1h values measured during daylight hours.

The WRF-Chem settings, including an introduction to the tagging approach and a summary of the model evaluation for NO, $NO_2$ and $O_3$, are presented in section 2. Model evaluation is discussed in detail in the supplementary material. Results for the contributions of UK and European precursor emissions, along with transport across the lateral model boundaries to surface $O_3$ in the UK are presented and discussed in section 3. Section 4 summarizes our findings.

## 2    Methods

We used the Weather Research and Forecasting model (WRF) version 3.7.1 (Powers et al., 2017) coupled with chemistry (WRF-Chem) (Grell et al., 2005). The model domain was centred at 3° E and 53° N, covering most of Europe, as shown in Fig. 1a. The spatial resolution was set to 27 km × 27 km, with 35 vertical levels starting from the surface up to 10 hPa.

The initial and boundary conditions (IC and BC, respectively) for meteorology were obtained from the ERA-Interim reanalysis dataset (Dee et al., 2011), which has a spatial grid resolution of 0.75° × 0.75° and 6-hour temporal resolution. IC and BC for

the chemistry fields were extracted from global simulations produced by the Chemistry Transport Model for $O_3$ and Related Chemical Tracers MOZART-4 GEOS-5 (Emmons et al., 2010). BCs were ingested into the model every 3 hours. The schemes used to parameterize the atmospheric processes are listed in Table 1. These are the same schemes deployed in (Mar et al., 2016) to evaluate meteorology, $O_3$, and NOx fields in a European domain using the MOZART-4 chemical mechanism.

Simulations were conducted for the period between April 24 and August 31 of 2015 for gas-phase chemistry using a tagged-

ozone mechanism based on the MOZART-4 chemical scheme. Note that omission of heterogeneous chemistry can lead to overestimation of $NO_2$ due to the absence of aerosol nitrate formation through the reaction of $OH + NO_2$ as well as $N_2O_5$ hydrolysis, which represents an important sink for $NO_2$ (Badia and Jorba, 2014; Archer-Nicholls et al., 2014; Stone et al., 2014). The first week of output was treated as model spin-up and hence discarded. The meteorology was not nudged but re-started every three days as in the methodology adopted in the second phase of the Air Quality Model Evaluation International

initiative (AQMEII) (e.g., Im et al., 2014). This decision was made after a test analysis showed that nudging of winds above the planetary boundary layer (PBL) and temperature at all layers, as done in Mar et al. (2016), leads to a representation of hourly $NO_2$ and $O_3$ mixing ratios in the East Anglia region (East of UK) that was inconsistent with observations. The nudging simulation predicted shallower boundary layers compare with that obtained using the re-starting method, particularly over the Norfolk Sea coast, leading to high concentrations of $NO_2$, especially at night time, and larger $O_3$ lost due to increased dry

deposition. Anthropogenic emissions of carbon monoxide (CO), NOx, sulphur dioxide ($SO_2$) and total NMVOCs for the European domain, including shipping lanes, were taken from the TNO-MACC-III European inventory (Kuenen et al., 2014) for the year 2011. The emissions were provided as yearly totals (kg yr$^{-1}$) by source sector following the SNAP (selected nomenclature for sources of air pollution) convention at a 0.125° × 0.0625° longitude-latitude resolution. For the UK domain, emissions were taken from the UK national emissions inventory (NAEI) for the year 2014, http://naei.beis.gov.uk/, which has

a spatial resolution of 1 km × 1 km. Biogenic emissions were calculated online using the Model of Emissions of Gases and Aerosols from Nature (MEGAN) version V2.04 (Guenther et al., 2012).

### 2.1  $O_3$ tagging mechanism

The contribution of hemispheric $O_3$ and domestic and European anthropogenic emissions to tropospheric $O_3$ in the UK is studied using the $O_3$ tagging technique developed in Lupaşcu and Butler (2019), in which $O_3$ molecules are labelled according

to the identity of their source regions. This is achieved by tagging NOx emissions at selected source regions and tracking them

through the formation of $O_3$, including the recycling of NOx via the production of odd nitrogen species (e.g., peroxyacetyl nitrate (PAN), nitric acid ($HNO_3$), and organic nitrates).

To implement the tagging method, a new chemical mechanism was created, "mozart_tag_kpp (chemopt=113)," containing the original chemical reactions in the MOZART-4 mechanism plus a duplicated set of reactions with additional tracers accounting for the source regions of interest. See Appendix A in Lupaşcu and Butler (2019) for a list of the model's edits to accommodate the new mechanism in WRF-Chem. The present study uses different sources and receptor regions. Furthermore, it does not attribute the contributions from the lateral boundary to any specific geographical location (e.g., the Force on Hemispheric Transport of Air Pollution, HTAP regions) as in Lupaşcu and Butler, (2019). Instead, the lateral boundary is tagged as a single source. $O_3$ formation requires both NOx and peroxy radicals from VOCs. Several tagging methods exist that can take different approaches to estimate the attribution of $O_3$ to these two chemically distinct precursors (Butler et al., 2018). Butler et al. (2020) demonstrates that anthropogenic NMVOCs emissions play a marginal role in regional scale $O_3$ production, with methane and biogenic VOCs being the most relevant chemical species for $O_3$ production. As the present study primarily focuses on the anthropogenic influence on $O_3$, the use of NOx tagging for $O_3$ source attribution is considered appropriate.

## 2.2 Receptors and source regions

Table 2 lists the tagged sources and receptor regions also highlighted in Fig. 1a and b, respectively. The chemical lateral boundary is defined as an independent source region (LB) and is provided by the MOZART-4 GEOS-5 model. Note that the LB tagged-region also includes $O_3$ contributions of stratospheric origin. Moreover, all $O_3$ that enters the model domain through the lateral boundaries is tagged as LB, and there is no feedback between the global model and WRF-Chem.

To generate the receptor regions, the UK domain is divided into its 12 administrative regions as used in previous air quality studies such as Heal et al. (2013), as shown in Fig. 1b: East Anglia, South-East, London area, South-West, Wales, West Midlands, East Midlands, Yorkshire and Humberside, North-East, North-West, Northern Ireland and Scotland.

The UK is well known for the regional variability of its weather. Generally, places in the east and south tend to be drier, warmer, sunnier, and less windy than those in the west and north (Jenkin, 2008). Thus, dividing the UK into several regions also serves to separate relevant meteorological features such as temperature, sunshine, precipitation, and wind, as well as emissions within each region that are known to have an impact on the spatial distribution and temporal variation of air pollutants such as $O_3$ (Coyle et al., 2002; Jenkin, 2008).

## 2.3 $O_3$ metrics for source contribution assessment

Current European and National air quality standards to mitigate the effects of $O_3$ on human health are expressed as 8 h averages. The regulatory framework establishes that the maximum 8 h mean $O_3$ concentration (MDA8) should not exceed 120 ug m$^{-3}$ (~60 ppbv) in the European Union (EU), and 100 ug m$^{-3}$ (~50 ppbv) in the UK. Here, the contributions to these health metrics were estimated by computing an 8 h moving mean of $O_3$ for each receptor region and selecting the days when the MDA8 exceeds 50 and 60 ppbv between May to August 2015. Once these were identified, tagged $O_3$ concentrations were extracted

for the same periods, and used in the analysis. The AOT40 is defined as the accumulated excess of hourly $O_3$ concentrations above 40 ppbv measured during daylight hours (between 08:00 and 20:00) Central European Time (CET) over a typical three-month growing season May-July. Here, the contribution of tagged $O_3$ to the cumulative metric AOT40 was calculated as the sum of the difference between hourly mixing ratios when $O_3$ exceeded the 40 ppbv threshold and 40 ppbv between 08:00 and 20:00 central European time (CET) from May-July over the most relevant arable farming areas in the UK, East Anglia and the South East, see Eq. (1). The target value in the EU and UK is 9000 ppb h$^{-1}$ (~18000 µg m$^{-3}$ hours) over a typical three-month growing season (May-July) averaged over 5 years.

$$AOT40= \sum_{i=l}^{90d} \sum_{h=8}^{20} \max (O_{3i,h} - 40, 0) ,  \tag{1}$$

## 2.4 Model evaluation: NO, NO$_2$ and O$_3$

Observational data were taken from the UK's Met Office Integrated Data Archive System and the European Monitoring and Evaluation Programme (EMEP). The EMEP air quality monitoring network records hourly measurements at regional background sites, mostly in farmland and rural areas (Tørseth et al., 2012). The choice to only analyze background representative stations is based on the need to provide an evaluation with spatial scales consistent with the model resolution. Kuik et al. (2016), for example, has shown that a 15 km resolution is too coarse to resolve the differences between urban and rural atmospheric chemical composition. The resolution of the domain considered here is even coarser. The EMEP network was therefore selected as it provides surface measurements at sites intended to represent regional background pollution.

Model evaluation is detailed in the supplementary material. Table S.2 summarises the domain-wide statistical performance for NO, NO$_2$ and O$_3$. The predicted temporal correlation coefficient ($r$) for NO and NO$_2$ is fairly low (0.3), which is a feature exhibited also in other regional studies in Europe using WRF-Chem e.g., Tuccella et al. (2012), Pirovano et al. (2012) and Lupaşcu et a. (2022). The model underestimates NO mixing ratios in most analyzed sites with a domain-wide MB of -0.3 ppbv. NO$_2$ mixing ratios, on the other hand, are generally overestimated with a domain-wide MB of 0.31 ppbv, and no specific patterns distinguished in the bias distribution. This is consistent with the negative NO and positive NO$_2$ biases obtained across Europe using MOZART-4 chemistry reported in Mar et al. (2016).

The model's temporal variation in hourly $O_3$ concentrations at most sites is well represented, with an average $r$ value of 0.6. The model tends to underestimate concentrations in most locations, with a domain-wide mean bias of -3.7 µg m$^{-3}$. Correlation values above 0.5 are obtained in most sites, particularly in the UK, see Fig. S4a in the supplementary material. In contrast, low $r$ values (~0.4) are concentrated on high-altitude sites, which might indicate difficulties in the model representing $O_3$ transport. This is in line with previous studies using MOZART-4 chemistry, such as Knote et., (2014), showing low production of peroxyacetyl nitrates (PAN), an essential reservoir for NO$_2$ and a key player in remote $O_3$ production. Correlation values are consistent with summer time $O_3$ values below 0.40 reported on the WRF-Chem model evaluation over a European domain on Mar et al. (2016) using MOZART-4 chemistry.

Fig. 2 shows that the day-to-day variation in hourly O$_3$ mixing ratios is well represented by the model, except for large under-predictions during 1–3 July and 22-24 August, particularly at stations on the east coast, e.g., Weybourne. Note that the observed maximum hourly O$_3$ at this site is larger than those seen inland, e.g., Lillington Heath and Harwell (2015). This may indicate inflow of O$_3$ and precursors from nearby large metropolitan areas within the UK (e.g., London) or to longer-range transport from continental Europe. Thus, underestimation of O$_3$ during those days may be caused by uncertainties in O$_3$ transport. This feature has also been identified in other source apportionment studies such as Lupaşcu and Butler. (2019).

## 3. Results and discussion

### 3.1 Contributions from tagged sources

#### 3.1.1 Spatial distribution and temporal variation

Consistent with previous work (e.g, Karamchandani et al., 2017; Lupaşcu and Butler, 2019; Butler et al., 2020), the hemispheric O$_3$ level, represented here by the LB source region, dominated the monthly O$_3$ concentrations in the UK during the entire study period with a mean relative contribution of ~71%, exhibiting a maximum in May (mean 76%), a minimum in June (mean 66%), and an increase in August (mean 72%); see Fig. 3. The mean contribution from the Eu super-region (FRA, GER, NET, LUX, BEL, NOS, Rest_CEu and Rest_Eu) accounts for nearly 16 % of the simulated monthly mean O$_3$. The largest Eu super-region contributions are observed in the UK locations closer to continental Europe and that together contain about 40% of the UK population (East Anglia, London area, South-East England and Yorkshire). The smallest Eu super-region contributions are observed over Scotland (May and June) and Ireland (July and August). Emissions from UK sources, on the other hand, accounted for about 13 % of the simulated monthly O$_3$. The domestic contributions tend to increase in June and decrease again in August. This monthly variation in the O$_3$ contributions is mainly caused by larger photochemical activity taking place during the summer months (e.g., Monks, 2005). Under these conditions, O$_3$ is formed by reactions involving the oxidation of NMVOCs in the presence of NOx and under the influence of solar radiation.

The spatial distribution of the monthly O$_3$ concentrations and the absolute contribution of the source regions (UK, LB and Eu super-region) to surface O$_3$ in the UK are shown in Fig. 4. The first column shows that the monthly surface O$_3$ concentrations are higher in May than during the summer months, in particular over Ireland, most of the Atlantic Ocean, north of UK, Scandinavia, and the northern North Sea. This is consistent with the Northern Hemisphere mid-latitude spring maximum (Monks, 2000), which is characteristic of remote locations and attributed to both transport of O$_3$ from the stratosphere to the troposphere (Monks, 2000) and transport of O$_3$ produced from anthropogenically emitted precursors (NOx and VOCs) (Monks, 2000; Butler et al., 2018). By contrast, the south east of the UK, southern North Sea and continental Europe exhibit sustained high O$_3$ mixing ratios throughout the entire analyzed period (May to August) reflecting a spring–summer maximum that is frequently attributed to photochemical O$_3$ production (Monks, 2000).

A marked latitudinal gradient is observed in the monthly $O_3$ mixing ratios, and in particular during June, July and August. Over the ocean areas, $O_3$ concentrations tend to be higher at the lower latitudes and in the North Sea, see Figs. 4e, 4i and 4m. Mean $O_3$ mixing ratios as low as 22 ppbv (Fig. 4i) are observed in most of the UK and Scandinavia in July while mixing ratios as high as 32 ppbv (Fig. 4e) are calculated for southern locations in the UK and Western Europe in June. This is consistent with previous estimates such as those in Butler et al. (2018). Part of the latitudinal gradient in surface $O_3$ over land can be attributed to the changing $O_3$ mixing ratios arriving from the Atlantic Ocean. Moreover, high mixing ratios in the south-east UK during summertime are generally associated with photochemical production of $O_3$ (Monks et al., 2005), in particular from anthropogenic NOx and biogenic NMVOCs emissions (Atkinson, 2000; Butler et al., 2018) as well as transport of $O_3$ rich air masses from continental Europe during anticyclonic conditions ( Jenkin et al., 2002; Lee et al., 2006; Francis et al., 2011; Romero-Alvarez et al., 2022). Low mean $O_3$ mixing ratios (as low as 20 ppbv), on the other hand, are observed in the vicinity of the main urban centres e.g., Greater Manchester, the Midlands and the London area (First column Fig. 4). This is because strong titration by excessive local NOx emissions takes place over the main urban centres, whereas high $O_3$ production rates are expected in the outskirts following the progressive reduction in NOx concentration relative to that of NMVOCs (Jenkin, 2008). Note that the latitudinal gradient across the UK is not evident in high altitude areas in Wales and northern England, having relatively high concentrations of $O_3$. This is because high altitude areas are much of the time above the shallow boundary layers that form over the lower lying land experiencing therefore larger exposure to $O_3$.

A suitable way to identify the areas influenced by fresh NOx emissions is comparing the mixing ratios of $O_3$ and Ox (= $O_3$ + $NO_2$). Ox is considered a conservative quantity as it is, to a large extent, free from the titration effect of $NO + O_3 \rightarrow NO_2$ (Kley et al., 1994). The effect of titration for July is evident in Fig. 5. When Ox is considered, the mixing ratios tend to increase over main urban centres such as London, Birmingham, Nottingham, Sheffield, and Greater Manchester, and to a lesser extent over Edinburgh and Glasgow. The NO titration effect is also observed within the main urban centres in continental Europe and along the shipping lanes over the North Sea and the English Channel due to the high NO content of ship emissions compared with that from NMVOCs (Aulinger et al., 2016).

The decrease in the monthly average $O_3$ mixing ratios towards the summer months over the Atlantic Ocean and most of the British Isles coincides with a progressive reduction in the contribution from LB $O_3$ over the same areas (second column in Fig. 4). Over remote marine areas, it is likely that the decrease in total $O_3$ is due to an increase in the photochemical activity and concentration of water vapor during the summer months. $O_3$ concentrations over land, on the other hand, are likely to be altered by both the changing background contribution from over the ocean, and by processes occurring at the regional and local scale (Jenkin, 2008). Such processes include $O_3$ scavenging near emission sources, changes in meteorology (wind direction influencing transport, and temperature and radiation influencing photochemical production of $O_3$), and planetary boundary layer stability (influencing vertical mixing and deposition) (AQEG, 2009). In addition, LB $O_3$ can be chemically lost near emissions sources, e.g., the Midlands and London area, shipping lanes and over an extended area on the southern part of the Atlantic Ocean, as shown in the map of net midday (11:00–14:00 UTC) near surface LB $O_3$ chemical production rate in Fig. 6. The figure also shows how the absolute contribution from the LB decreases southward and eastwards. Over the Atlantic,

part of this can be attributed to a greater chemical $O_3$ sink due to the increase in photolysis of $O_3$ and subsequent production of OH radicals from water vapor (Johnson et al., 1999). Transport of $O_3$ from the stratosphere might also influence the spatial gradient in the contributions.

A marked reduction of LB $O_3$ is observed over the UK in column 2 Fig. 4 (e.g., a decrease of ~ 10 ppbv between LB $O_3$ over the ocean and the UK). Depletion of surface $O_3$ by dry deposition, and chemical loss processes within the UK, such as the reaction of $O_3$ with NO, may help explain the observed spatial gradient. Reductions in LB $O_3$ due to the effect of local $O_3$ scavenging by reaction with NO in the urban centres as illustrated in in Fig. 7 might be an additional caused.

Whereas the absolute contribution from the LB over the UK tends to decline with distance towards the south-east, the absolute contribution of UK anthropogenic NOx emissions to the mean surface $O_3$ over the UK (Fig. 4 third column) decreases from the south-east to the north and west. The mean contribution of the UK-to-UK surface $O_3$ concentrations is marginal. Over most of the north and the west, the UK contributions ranged from 1–3 ppbv. By contrast, maximum UK contributions can reach up to 7 ppb in the east and the Midlands during the summer months. These areas tend to be drier, warmer, and sunnier than those regions further west and north, features that are conducive to photochemical $O_3$ formation (e.g., Coyle et al., 2002; Jenkin, 2008). Furthermore, these regions contain some of the UK largest cities (e.g., London, Birmingham, Nottingham, Manchester and Leeds) which can lead to net $O_3$ formation downwind of the emission sources where the NOx titration effect is reduced. Indeed, overall, the south and east of the UK exhibit the highest midday (11:00–14:00) $O_3$ chemical production from UK anthropogenic sources, see Fig. 8a. Lower $O_3$ chemical production is instead observed in the west and the north beyond Yorkshire and Humberside, as shown in Fig. 8a. On the other hand, the UK makes a positive contribution to $O_3$, of around 4-8 ppbv, downwind over continental Europe.

The contribution from European NOx emissions to the mean surface $O_3$ over the UK (Fig. 4 fourth column) is comparable to that observed from the UK contribution and tends to be higher along much of the eastern, southern and south-west borders, reaching up to 10 ppbv in East Anglia during July. This reflects the effective transport of continental $O_3$ by south-easterly winds during high $O_3$ pollution events. The European contributions then decrease towards the northern and western areas of the UK, with a minimum (1–3 ppbv) over Scotland and Ireland. Fig. 8b demonstrates that surface $O_3$ from anthropogenic sources from the Eu super-region is mainly produced outside the UK. This indicates that the contribution from EU emissions to UK surface $O_3$ is predominantly due to transport of $O_3$ rather than its NOy precursors. Also, $O_3$ from EU sources is chemically lost nearby the largest cities in UK (e.g., London area, Birmingham, Nottingham, Manchester and Leeds) and in the English Channel and North Sea, as shown in the plot of net midday, surface chemical $O_3$ production rate from European anthropogenic NOx emissions in Fig. 8b. Chemical production is generally concentrated over central Europe and the Baltic Sea. By contrast, chemical loss happened within the main urban centres, nearby point sources and along the shipping routes around Western Europe, the North Sea and English Channel, e.g., sites previously identified to be influenced by NOx titration.

### 3.1.2      Regional dependence

The modelled contributions of the different source regions to the UK receptor regions for May are presented in Fig. 9. The
figure contains 12 nested pie charts, each one associated with a receptor region in the UK that shows the absolute and relative
contributions to $O_3$ mixing ratios in the UK from all anthropogenic NOx sources, including ship emissions. Note that the
contributions from the Rest_Eu source need to be carefully interpreted since these include emissions from the Republic of
Ireland, Iberian Peninsula, Southern EU, South-eastern EU, Eastern EU, Northern EU, and ship emissions from the Atlantic,
Baltic Sea and the Mediterranean.

The LB is the principal contributor to the modelled mean $O_3$ mixing ratios in every receptor region. The contributions peak in
May (mean absolute contribution 25 ppbv), reflecting the seasonal cycling in the northern hemispheric background $O_3$ (e.g.,
Monks, 2000; AQEG, 2009). Contributions from this source are more prominent in the regions located in the north, east, and
north-west of the UK, e.g., Scotland (30 ppbv), Northern Ireland (28 ppbv), North-East (27 ppbv), the North-West, and Wales
(26 ppbv). These regions contain about 20% of UK population and are primarily impacted by westerly flows and associated
hemispheric $O_3$ background due to their geographical position (AQEG, 2009). Also, they generally experience less than 10
days with $O_3$ concentrations above the EU limit of 120 μg m$^{-3}$ (DEFRA, 2020) because of low NOx emissions locally. The
contributions from the LB source in the South-East, East Anglia and East Midlands, on the other hand, can be up to 8 ppbv
smaller than in the East of UK, particularly during summer time, see figures S.10 - S.11 in the Section S.2 in the supplementary
material.

The UK contributions are generally more significant in the east, south-east, and the Midlands, showing a maximum value in
June and July in every receptor area, Figures S.10 and S.11 in the supplemental material. The source region provides up to
20% of the surface $O_3$ mixing ratios in East Anglia, 18% in the London area and East Midlands, and 16% in Yorkshire and the
South East, making it the second-biggest source of $O_3$ in these locations after the LB. This area incorporates about 50% of UK
population and often experiences more than 10 days with $O_3$ concentrations above the EU and UK $O_3$ threshold (concentration
> 120 and 100 μg m$^{-3}$, respectively) (DEFRA, 2020). The Eu super-region, on the other hand, is the second-largest source
region in the northern and western UK, with contributions in summertime reaching up to 10% in Scotland and 16% in South-
West England, and 14% in Wales. Regardless, this source region still significantly impacts the South-East, East Anglia, and
London, where the relative contributions can increase from 13, 12, and 13% in May to 16, 15, and 16% in July, respectively.
The contributions from ship emissions from the North Sea and English Channel are significantly lower than those from UK
sources and Eu super-region (3-4% of the total contribution in the Southeast and East Anglia in May and up to 6% of the total
surface $O_3$ during the summer months). The impact is also less important in the west than in the east and south of the UK due
to the proximity with the region. As for the relative contributions from the different Eu sub-regions (inner circle in Fig. 9),
these are largely influenced by the geographical situation of the receptors and the predominant wind direction. In every
receptor, the principal contributor from the Eu super-region is the Rest_Eu source region, providing between 60-70% of the
Eu super-region $O_3$ in May and up to 83% of the $O_3$ during summertime. The relative contributions of the Rest_Eu region is

larger in the northern and western locations, in particular during the summer months when there is a marked difference in the distribution of the contributions across the UK regions. The summer months see an increase in the input from France, Germany and the Benelux region, in particular during anticyclonic weather conditions and over the receptor regions located in the south and east of the UK (e.g., South East England, East Anglia, the London area and the East Midlands), see figures S.10 - S.11 in the supplementary material. This is consistent with results of studies on extreme $O_3$ in the EU and the UK reporting an increase in surface $O_3$ concentrations under anticyclonic conditions (e.g., Pope et al. (2016); Ordóñez et al. (2016); Romero-Alvarez et al. (2022)). Romero-Alvarez et al. (2022), in particular, has shown that a wide area of high pressure centred over the Netherlands coast affected most of England during the first days of July 2015. During the same period, regions such as the East Anglia reported increases in $O_3$ mixing ratios of up to 16.6 ppbv h$^{-1}$ that overlapped with wind direction changes from south-southwest to south-southeast.

Depending on the predominance of the wind direction (south- southeast and south-southwest), $O_3$ from anthropogenic sources within France can impact both the west and the east of the UK. The contribution is greater in the southern UK due to the proximity to the source region. The contributions from the Benelux region and Germany are more significant in the east of the UK due to the proximity with the continent and association with easterly flows (east and southeast), about 14% and 6% of the Eu super-region in the East Anglia during the summer months comes from these two source regions, respectively.

Notably, anticyclonic conditions and easterly winds in the UK have been associated with enhanced $O_3$ concentrations whereas cyclonic conditions and westerly winds have been linked to $O_3$ transport from the UK mainland and cleaner air from the North Atlantic (Jenkin et al., 2002; Pope et al., 2016; Romero-Alvarez et al., 2022). The contribution patterns described above may thus serve as predictors of future $O_3$ source apportionment over the UK regions.

## 3.2  Contributions to regulatory $O_3$ metrics

### 3.2.1    MDA8 $O_3$ exceeding 50 ppbv

The mean contribution from each source region for the hours when the MDA8 $O_3$ exceeded 50 ppbv at each receptor area from May to August is presented in Fig. 10. The figure shows large contributions from source regions that were not seen as dominant sources. France, for example, becomes a major source, particularly in receptors in densely populated areas such as the south and east of the UK. The absolute mean contributions at the sites sometimes exceed the input from the LB $O_3$ (mean value ranging between 10 and 15 ppbv, and maximum reaching up to 35 ppbv in the London area). The impact of UK NOx on $O_3$ varies across the sites, but in general, its share increased from the southeast to the north. In the Midlands, the North East, North West, Scotland, and Yorkshire, $O_3$ from UK sources becomes dominant, surpassing the LB mean input in most receptors. In the remaining locations, the UK source is the third-largest input for surface $O_3$ except for the South West where most of the $O_3$ comes from France (mean ~18 ppbv), the LB (mean ~14 ppbv), NOS (mean ~6 ppbv) and Rest_Eu (mean ~8 ppbv). The impact from the shipping component (NOS) also becomes important in all receptor regions with an estimated mean of 4-7 ppbv. $O_3$ from Central Eu, Germany, Netherlands, Belgium and Luxembourg, on the other hand, is almost negligible in the

west of the UK (mean less than 1 ppbv). However, their impact increases towards the east and north with mean values ranging about 1-6 ppbv (e.g., in the East Midlands, North-East, Yorkshire and the Humberside and Scotland) reflecting the efficient transport of polluted-loaded air masses under anticyclonic conditions.

### 3.2.2  MDA8 $O_3$ exceeding 60 ppbv

Fig. 11 shows the mean contribution at each receptor area to hourly surface $O_3$ when the MDA8 $O_3$ exceeded the 60 ppbv thresholds. There were two occasions when the modelled MDA8 exceeded 60 ppb, the main occasion being the episode on the 1st of July. Romero-Alvarez et al. (2022) has shown that MDA8 $O_3$ above 50 ppbv in the Southeast and East Anglia regions in July 2015 coincided with days when easterly winds prevailed (east-southeast flows). In contrast, MDA8 $O_3$ above 60 ppbv coincided with a shift in the wind direction from east-southeast to south-southeast and south and a sharp rise in the surface temperature.

France was the most significant contributor to the build-up of $O_3$ when the mixing ratios exceeded the EU threshold in South East England (mean ~18 ppbv), East Anglia (mean ~21 ppbv), and the London area (mean ~26 ppbv) because convergence of westerly and south-easterly winds in the west of the UK diverted the contributions of domestic sources from these regions, as reported in Romero-Alvarez et al., (2022). $O_3$ from UK NOx emissions, on the other hand, has a greater impact on the East Midlands (mean ~16 ppbv), Yorkshire and Humberside (mean ~15 ppbv). In the South-East and the London area, the contributions from Rest_Eu equal those from UK $O_3$, while the influence is comparable to that from the west and Central Europe in the rest of the regions. As in the contributions to the MDA8 $O_3$ threshold of 50 ppbv above, the lateral boundary component remained nearly constant in all receptor areas with a mean contribution about 12 ppbv. This is because most of the UK's weather was dominated by anticyclonic conditions. The impacts from the North Sea and the English Channel are also important in all receptor regions, with a mean between 4-7 ppbv. Results suggest that ship emissions along these routes affect the air quality of the UK, particularly over the East and South-east. However, the current model configuration does not consider the chemical evolution of the different emitted species (chemical loss and production rates) during the dispersion of the ship plume. In fact, once species are emitted, they are instantaneously mixed in each model grid cell (27 x 27 km). In the case of chemically reactive species such as NOx, this can lead to overestimations of both NOx and $O_3$ concentrations due to the non-linearity of the chemical processes involving NOx and $O_3$ evolution during the dispersion of the ship plume (e.g., Huszar et al., 2010; Van Der Werf et al., 2010).

### 3.2.3  AOT40 index

The average simulated AOT40 for 2015 at the two most relevant arable areas in the UK: East Anglia and the South East is 3674 and 1833 µg m$^{-3}$ hours, respectively (the target value for the EU and UK is (~18000 µg m$^{-3}$ hours). Fig. 12 shows the source contributions to the surface $O_3$ when the mixing ratio exceeded 40 ppbv during the daytime hours (08:00 and 20:00) Central European time from May to July in two receptor regions in the UK. When exceedances to the hourly surface $O_3$ mixing

ratios above 40 ppbv is considered, the LB component becomes the dominant source in both receptor regions (estimated mean concentration between 21-24 ppbv) as its threshold is close to the tropospheric baseline ozone level associated with maritime Nort Atlantic air masses. The second largest contributor is the UK with a higher impact in the East Anglia region (estimated mean concentration 10 ppbv) than in the South East (estimated mean concentration 6 ppbv). The third, fourth and fifth contributions in East Anglia come from the Rest_Eu and Rest_CEu regions and France, while in the South East the contributions come from the North Sea and English Channel, Rest_Eu and Germany. The contributions from the Netherlands, Belgium and Luxembourg become almost negligible.

Note that the AOT40 metric assesses the impacts of $O_3$ on the vegetation by considering an $O_3$ threshold (e.g., concentrations above 40 ppbv) during the months when plant growth is most likely to be affected and when daytime $O_3$ concentrations are at their highest. However, research experiments have shown that the response of plants to $O_3$ exposure is non-linear due to a mismatch between the peak daytime $O_3$ concentrations and stomatal opening (Heath et al., 2009). This means that the effective amount of $O_3$ taken up by plants is not always correlated to the ambient $O_3$ concentrations. The AOT40 index does not account for the plant's physiological control of stomatal opening, which limits the potential of the index to accurately assess the impacts of $O_3$ on the vegetation. Future work should consider flux-based metrics which have proven to be more suitable for ozone-risk assessment on plants as they take into account the ambient concentration of $O_3$, the physiological control on stomatal openings, and the efficiency of leaf antioxidant system (Fares et al., 2010) as well as improving the $O_3$ deposition routines in WRF-Chem so that they can take into account factors such as stomatal opening cycles.

## 4    Conclusions

An $O_3$ tagging technique within the WRF-Chem model was used to investigate the origin of surface $O_3$ from May to August 2015 and the contribution of different source regions to $O_3$ regulatory metrics in the UK. Evaluation against observations presented in the supplementary material has shown that the model setup gives a good representation of $O_3$ in the European domain.

Domain wide examination demonstrates that the hemispheric $O_3$, here represented by lateral boundary $O_3$, has the largest impact on the concentrations of $O_3$ in the UK, with an estimated 71% of the modelled monthly mean surface $O_3$ coming from this source region. About 16 % of modelled surface $O_3$ is produced from anthropogenic NOx emissions within the EU that contain lumped NOx emissions from continental Europe, the Republic of Ireland and from ship emissions in the Atlantic, North Sea, Baltic Sea and the Mediterranean. UK emissions (England, Scotland, Wales and Northern Ireland) contributed 13%.

Assessment of the contributions to different receptor regions in the UK revealed that the UK relative contribution to UK surface $O_3$ tends to be higher in June and July with a marked spatial gradient, with high mixing ratios obtained in the south-east and lower values in the north and west. In fact, UK NOx emissions are the second largest contributor to surface $O_3$ in the East Midlands, West Midlands, Yorkshire and the Humberside, East Anglia, South East England and the London area after the lateral boundary source region. The monthly and spatial variation of the contribution of UK NOx emissions to UK surface $O_3$

is primarily caused by larger photochemical activity taking place during the summer months in the south and downwind of emissions sources. Similarly, the absolute contribution from European sources to UK surface $O_3$ tends to be higher in June and July and along much of the eastern, southern and south-west borders reflecting the effective transport of continental $O_3$ by south-easterly winds during $O_3$ pollution events. The tagging technique also shows that $O_3$ from this region is generally transported to the UK rather than produced in-situ.

$O_3$ tagging has also made it possible to demonstrate that more stringent emission controls would be required in different source regions for compliance of UK and EU $O_3$ standards, e.g., MDA8 $O_3$ of 50 and 60 ppbv. Emissions controls in France, in particular, would significantly reduce $O_3$ concentrations in densely populated areas such as the South East, South West and East Anglia while domestic emissions controls are more relevant for the Midlands and the north of the UK. Exposure thresholds, such as those considered in the AOT40 $O_3$ metric, are instead most affected by lateral boundary components in first place followed by UK NOx emissions. Emissions controls in regions such as the East Midlands, West Midlands, Yorkshire and the Humberside, East Anglia, South East England and the London area will aid in the mitigation of the impacts on crops. Nonetheless, emission controls will be also necessary over the larger Northern hemisphere area.

The results from model simulation should be interpreted in the context of the observed bias in $O_3$ (domain mean bias (MB)= -3.71 $\mu g\ m^{-3}$) and the underestimation of the number of days with the MDA8 $O_3$ above 50 and 60 ppbv and AOT40 metric. Also, controlling emissions of NO would not necessarily translate into reduction of $O_3$ concentration in the UK. In fact, For the AOT40 measure in rural regions it seems that reducing UK emissions might well still help improve the situation whereas for urban regions reducing NOx will increase $O_3$ concentrations due to a reduction in the titration effect. In this regard, future work should consider extending the tagging mechanism to include the competing NOx–VOC interactions in $O_3$ production. Emission perturbation studies might also complement the investigation by adding an understanding of response of $O_3$ to different emission control scenarios.

**Code and data availability**

The WRF-Chem model is publicly available on http://www2.mmm.ucar.edu/wrf/users/download/get_source.html. The modification described in Section 2 as well as the model output are available online via Zenodo at https://doi.org/10.5281/zenodo.6968040 and https://doi.org/10.5281/zenodo.6968649.

**Author contributions**

JRA designed the study, conducted the numerical simulations and analyzed the data. The original tagging method was developed by AL and TB and adapted by JRA for this study. DL and SN developed the emissions pre-processor. JRA wrote the paper under the supervision of CR, SD and AB, with contribution from all authors.

**Competing interests**

The authors declare that they have no conflict of interest.

## Acknowledgements

The authors would like to thank Dr. Fabio Di Gioacchino and Ravan Ahmadov for providing feedback. We thank Leo Earl
and Jimmy Cross (high-performance computing UEA) for their support in the compilation of the WRF-Chem model. We thank
TNO for access to the TNO-MACC III emissions inventory. The WRF-Chem simulations were done on the high-performance
research computer of the University of East Anglia, UK.

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

**Table 1.** Parameterizations options used in the study

| Process | Scheme |
|---|---|
| Cloud microphysics | Lin et al. scheme (Lin et al., 1983) |
| Radiation (short wave) | RRTMG (Iacono et al., 2008) |
| Radiation (long wave) | Goddard shortwave scheme (Chou and Suarez, 1994) |
| Boundary layer physics | Yonsei University scheme (Hong et al., 2006) |
| Surface layer | MM5 similarity based on Monin–Obukhov scheme (Beljaars, 1995) |
| Land surface processes | Noah land surface model (Chen and Dudhia, 2001) |
| Cumulus convection | Grell 3-D scheme (Grell and Dévényi, 2002) |

**Table 2.** List of tagged source regions

| Source region | Abbr. | List of countries or source type |
|---|---|---|
| Hemispheric $O_3$ | LB | Lateral boundaries |
| France | FRA | France |
| Germany | GER | Germany |
| Netherlands | NET | Netherlands |
| Luxemburg | LUX | Luxemburg |
| Belgium | BEL | Belgium |
| North Sea and English Channel | NOS | North Sea and English Chanel |
| UK | UK | England, Scotland, Wales and Northern Ireland |
| Rest of central Europe | Rest_CEu | Austria, Switzerland, the Czech Republic, Hungary, Poland, Slovakia, Slovenia and Romania |
| Rest of Europe | Rest_Eu | Remaining areas in the model domain including the Republic of Ireland, Iberian Peninsula, Southern Europe, South-eastern Europe, Eastern Europe, Northern Europe, and shipping emissions from the Atlantic, Baltic Sea and the Mediterranean. |




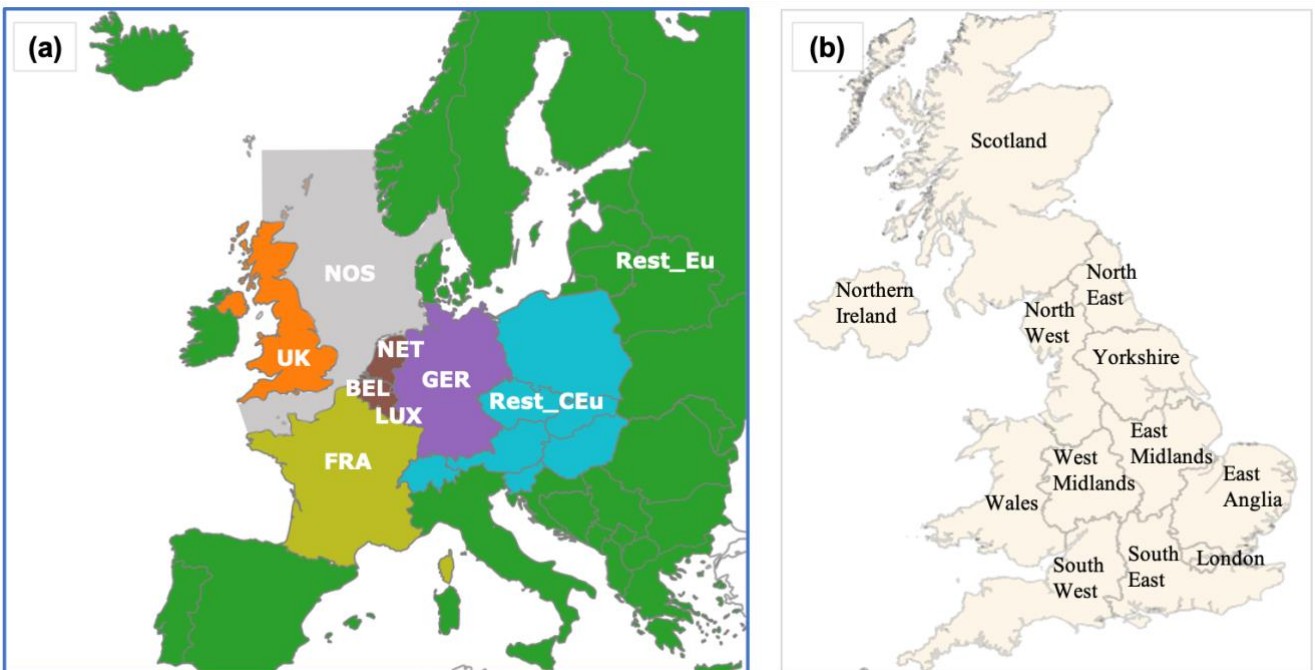

Figure 1. Source regions and receptors: (a) division of domain into 9 source regions. Note that Rest_Eu source region also includes ship emissions from the Atlantic, Mediterranean and Baltic Sea whilst emissions from shipping routes in the North Sea and the English Channel are tagged as NOS. The blue line surrounding the domain indicates LB tag. (b) map of the UK showing the receptor regions.




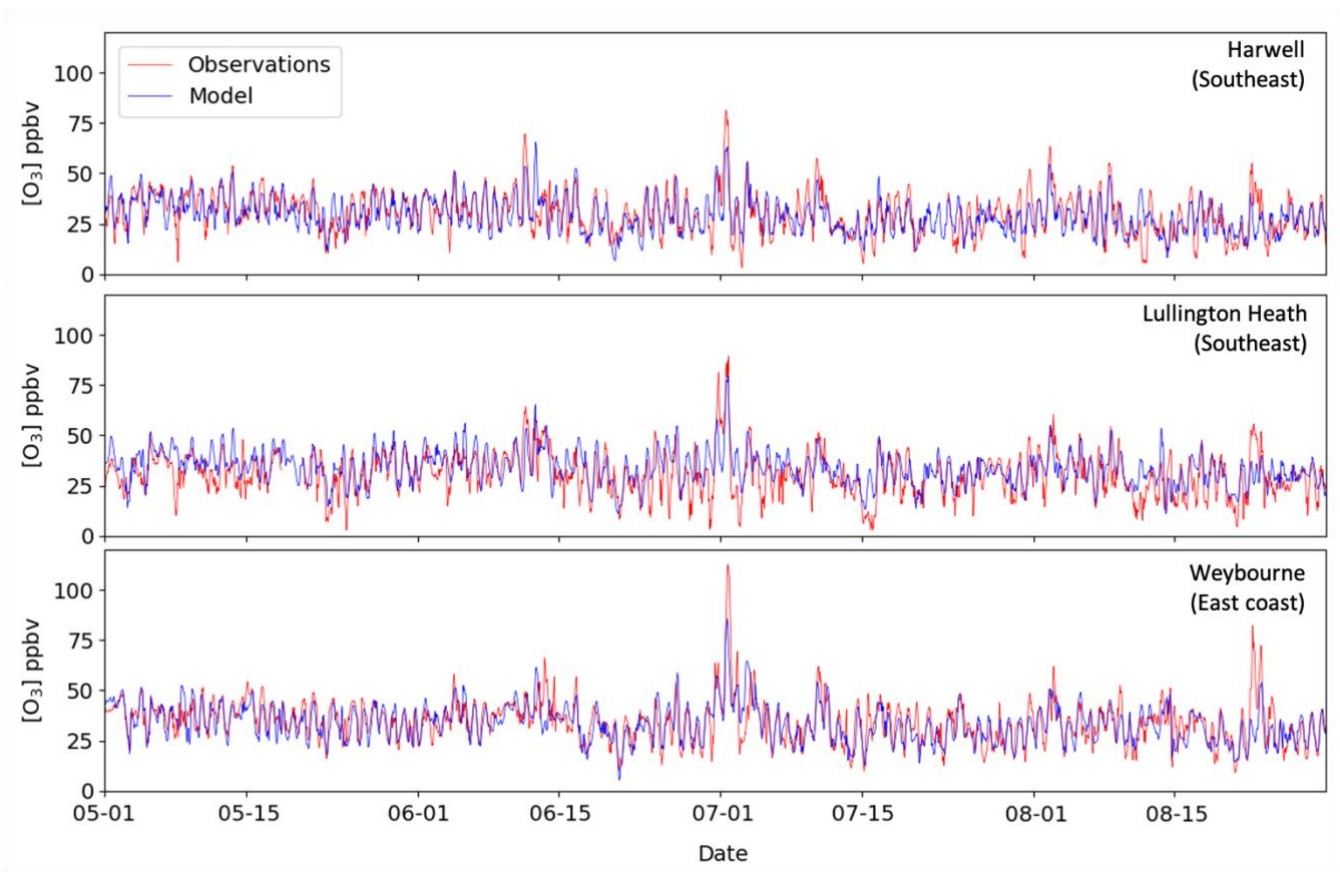

**Figure 2.** Modelled and observed hourly O$_3$ mixing ratios from May to August 2015 at three sites over the UK.




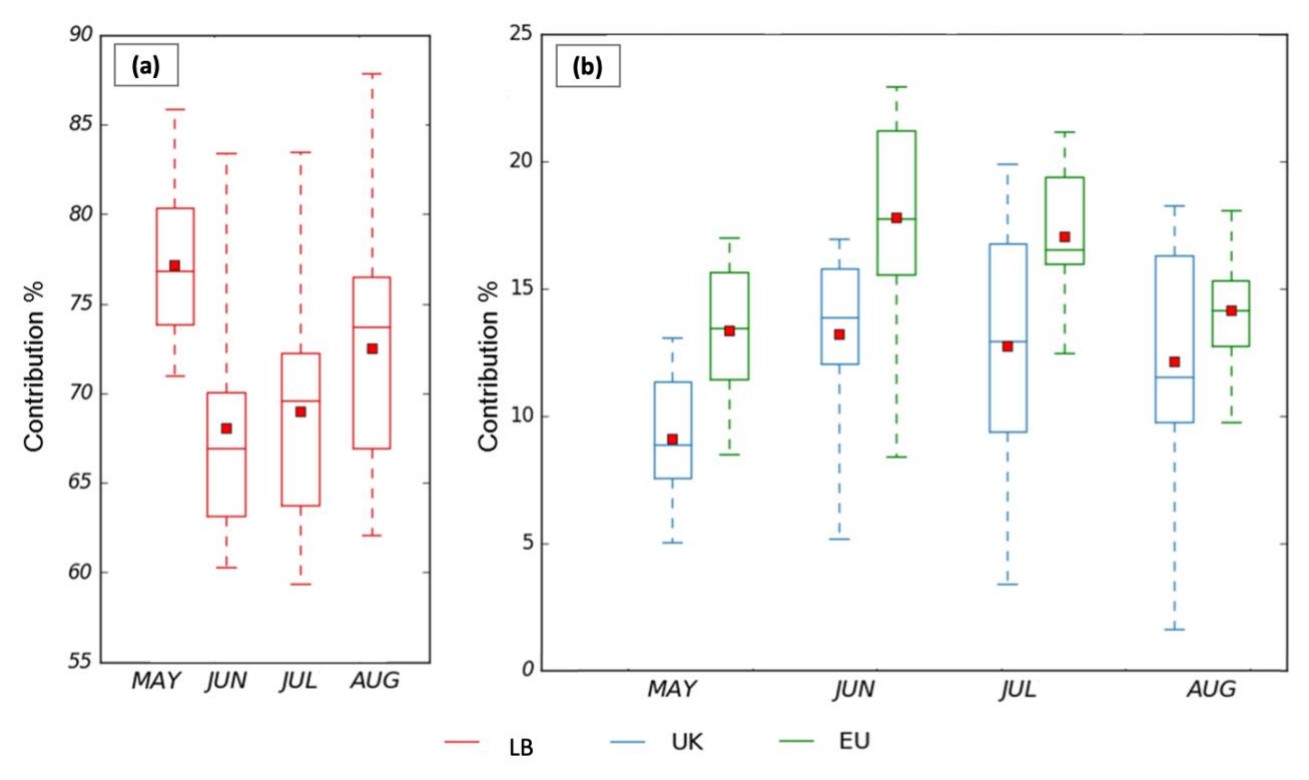

**Figure 3.** Monthly relative contributions (%) to surface $O_3$ in the UK from May to August 2015 from (a) the lateral boundaries (LB) and (b) the UK and the Eu super-region. The lower and upper end of the boxes indicate the $25^{th}$ and $75^{th}$ percentiles, the central bar the median, and the red square the mean. Whiskers indicate the maximum and minimum. Note the differences in the scale in the y axis.




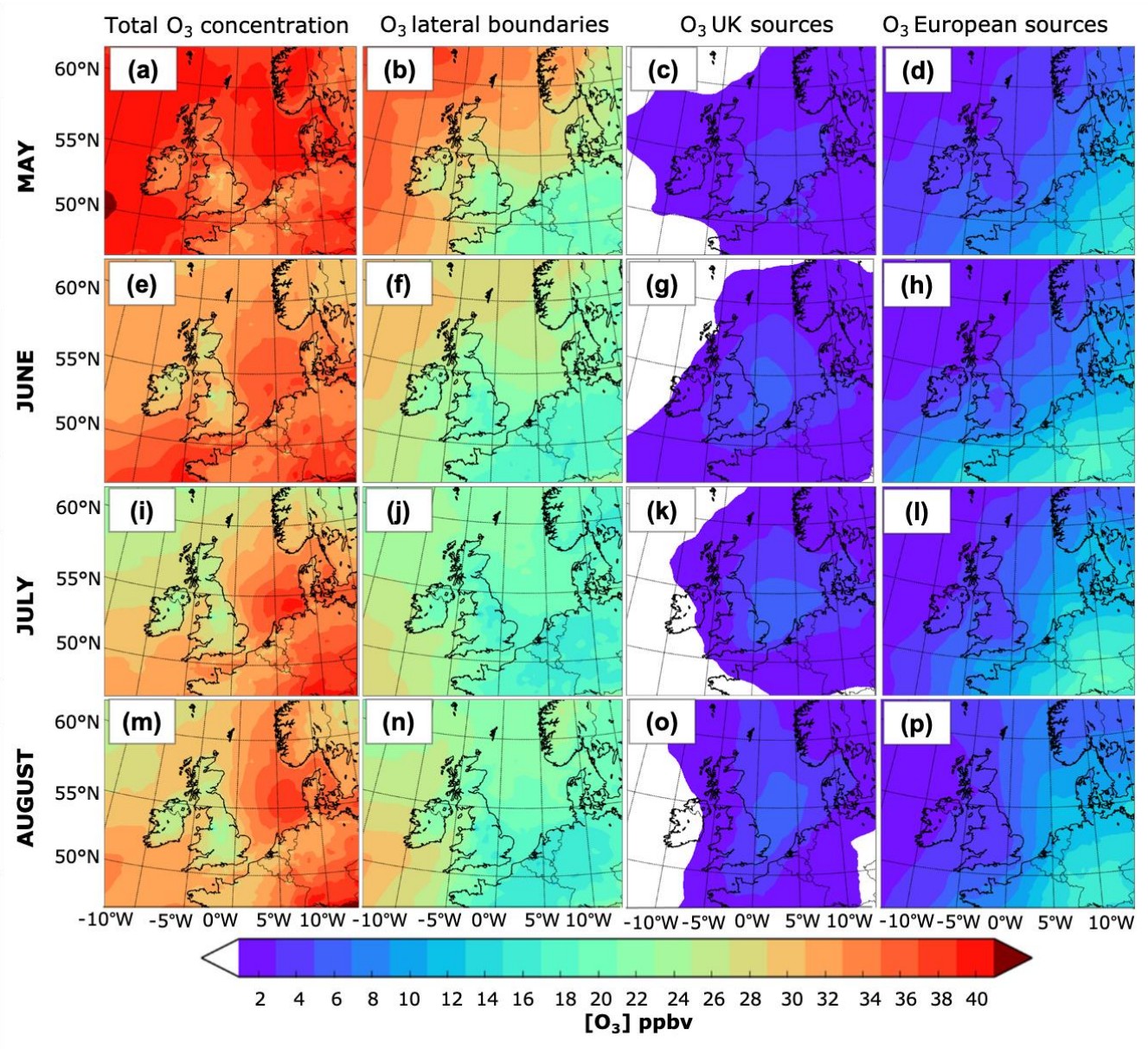

**Figure 4.** Spatial distribution of the monthly mean surface O₃ from May to August 2015. The first column depicts the mean O₃ mixing ratio in May, June, July and August; the absolute monthly contribution from the lateral boundaries is shown in the second column; the third column shows the contribution from UK emissions; and the contribution from the Eu super-region, which includes emissions from main shipping routes over the European seas and the Atlantic, is presented in the fourth column.

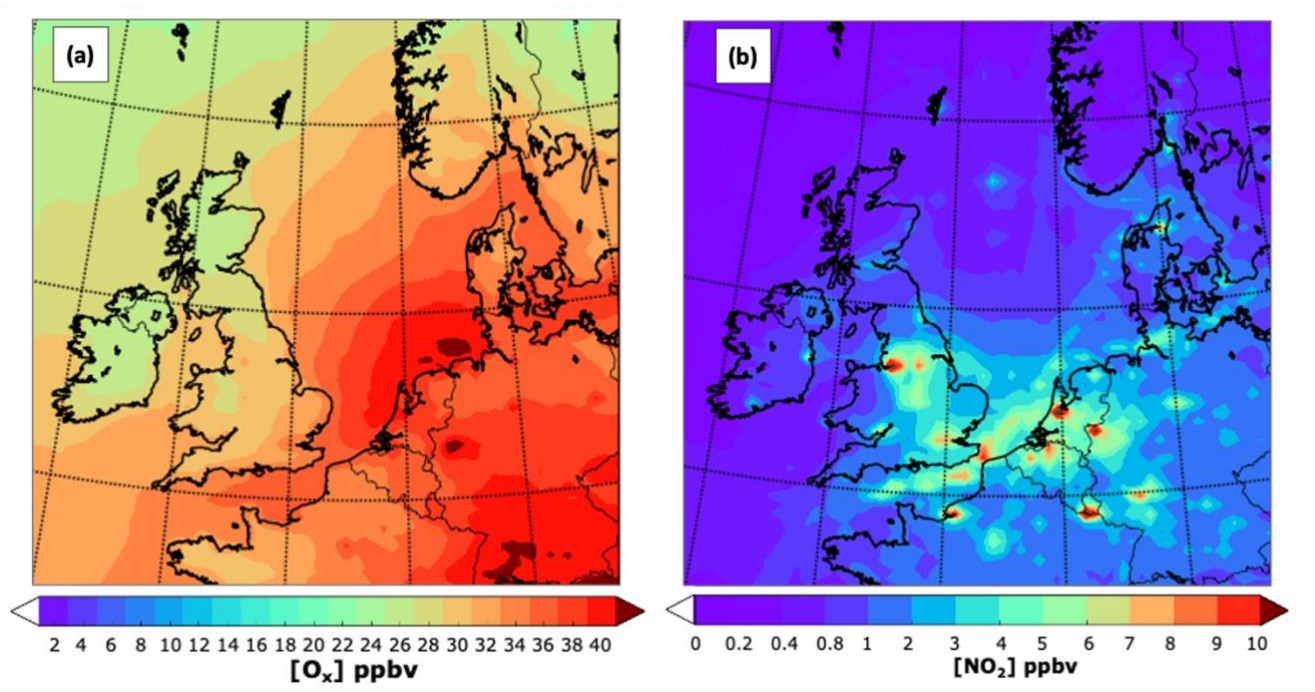

**Figure 5.** Close up of the spatial distribution for July 2015 of (a), mean $O_x$ mixing ratio ($O_3 + NO_2$) and (b) $NO_2$. Note the different scales.




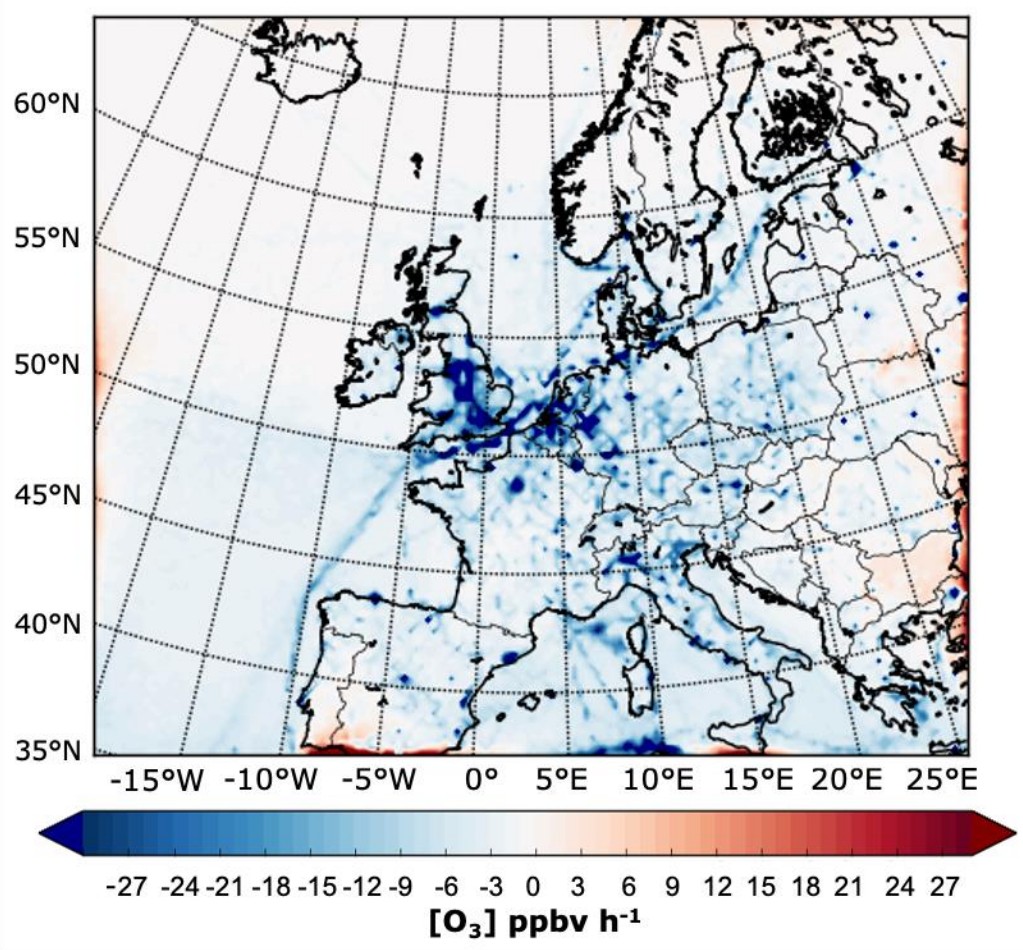

Figure 6. Net midday (11:00–14:00 UTC) near surface lateral boundary $O_3$ chemical production rate in ppbh$^{-1}$ on July 2015. Note that $O_3$ production is driven by tagged LB NO$_Y$ that has entered the model domain via the lateral boundaries.

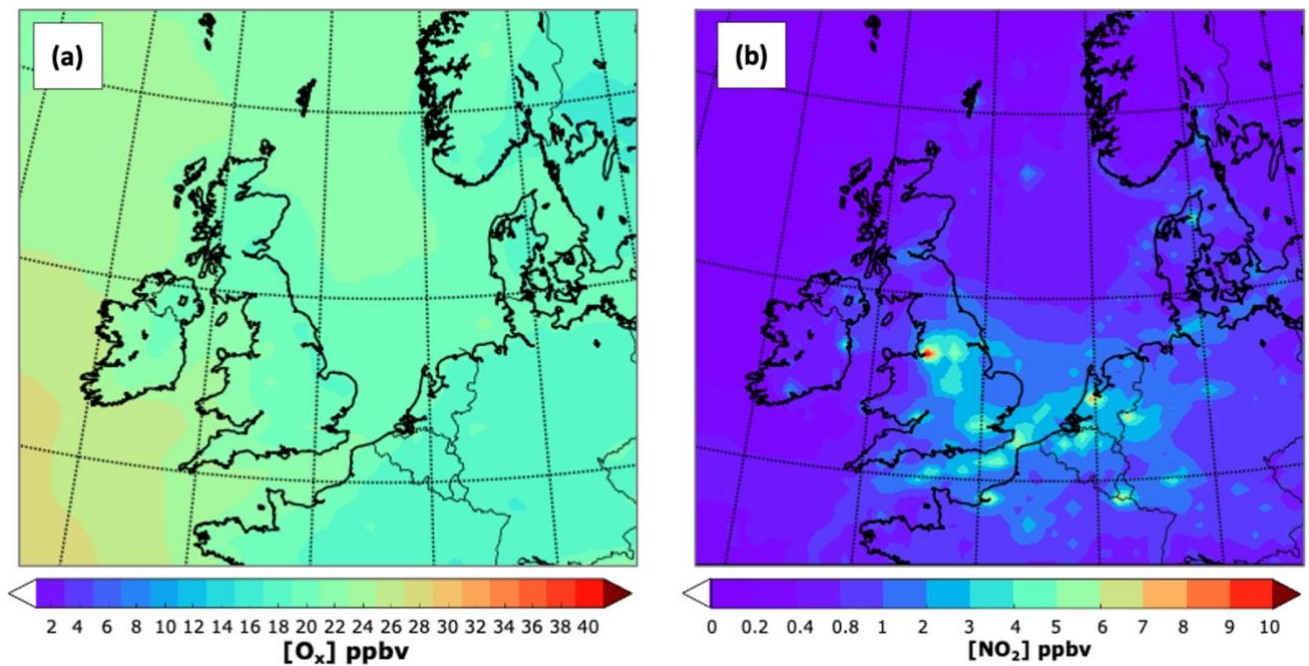

**Figure 7.** Close up of the spatial distribution for July 2015 of (a) lateral boundary mean $O_x$ mixing ratio ($O_3$ + $NO_2$), and (b) $NO_2$. Note the different scales in (b).



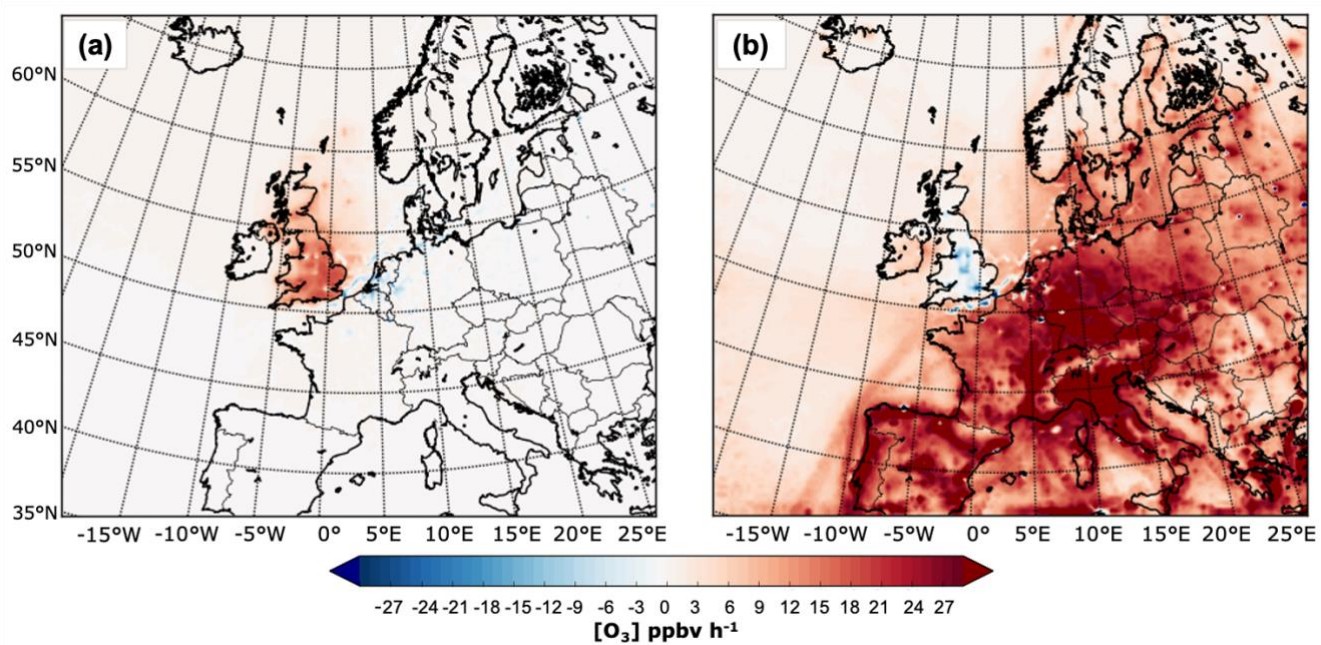


**Figure 8.** Net midday (11:00–14:00 UTC) surface chemical production rate in ppb h$^{-1}$ on July 2015 for O$_3$ from UK (a) and European anthropogenic NO$_x$ emissions (b).





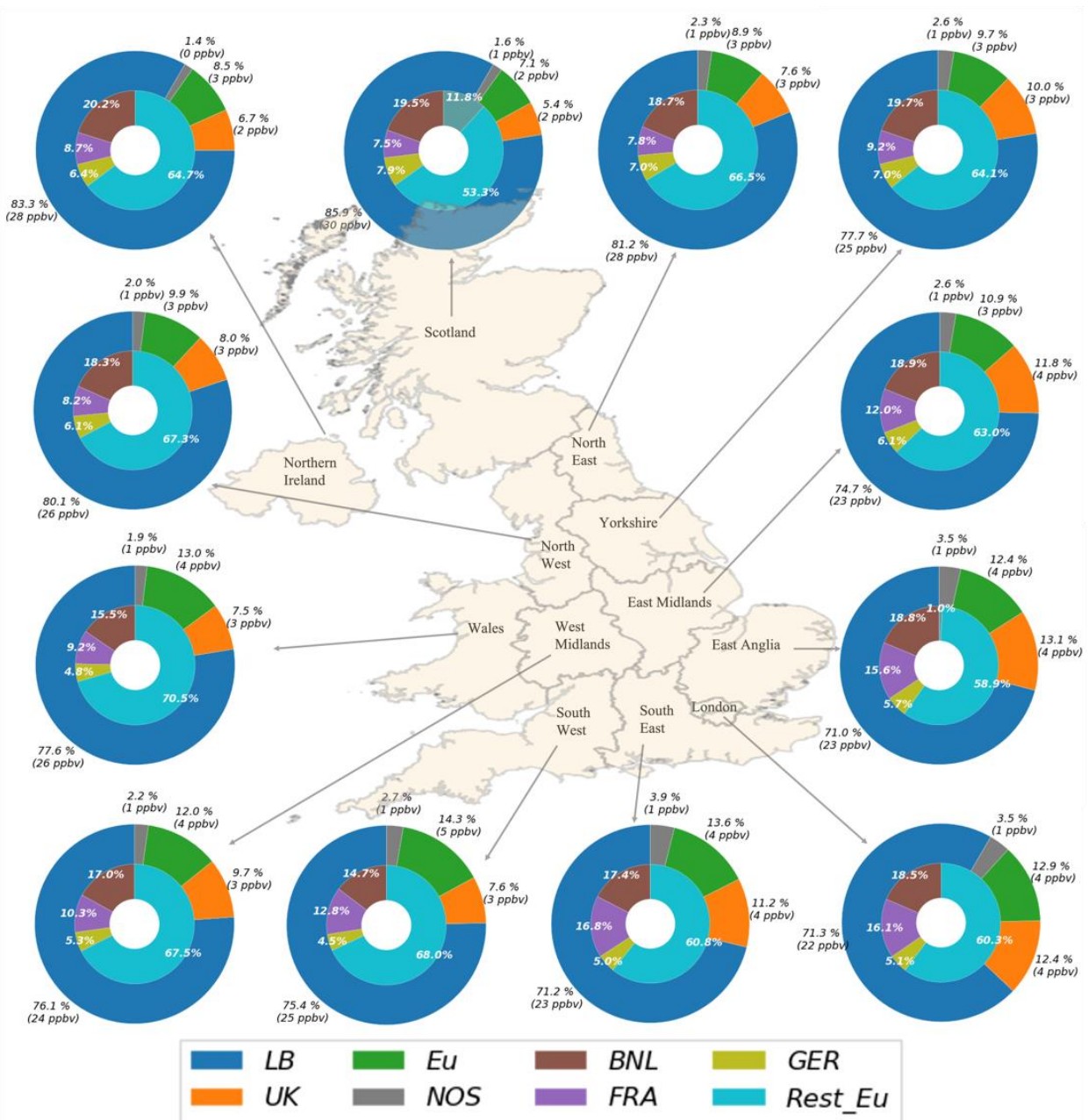

**Figure 9.** Simulated contributions to the mean O$_3$ mixing ratios in May 2015 over 12 receptors regions in the UK. Outer circle depicts the
contributions from LB, UK, Eu super-region (Eu), and the NOS. The inner circle breaks down the contribution from the Eu super-region
into four sub-regions: The Benelux (BNL), France (FRA), Germany (GER), and the rest of Europe (Rest_Eu). Note that the values correspond
to the contributions from anthropogenic sources only, with the exception of the LB which includes O$_3$ from stratospheric origin.

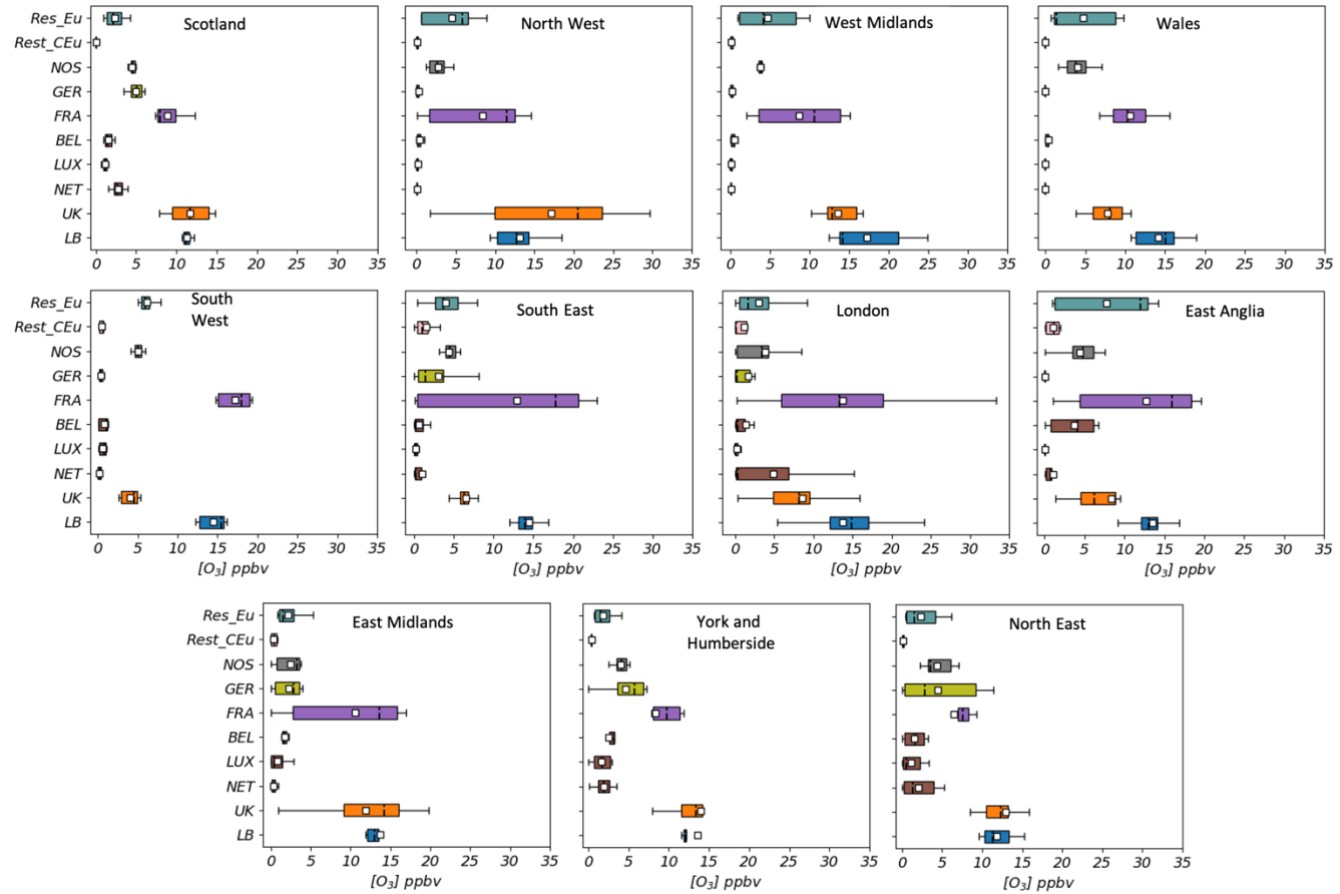

**Figure 10.** Hourly contributions, in ppbv, to surface $O_3$ at 11 UK receptor regions from 10 source regions (UK, background (LB), the Netherlands, Luxembourg, Belgium, France, Germany, rest of Central Europe (Rest_CEu), North Sea and English Channel (NOS) and rest of Eu) during days when the MDA8 is above 50 ppbv between May and August. The lower and upper ends of the boxes indicate the 25th and 75th percentiles, the black dashed line the median, the white boxes the mean and the whiskers the minima and maxima.

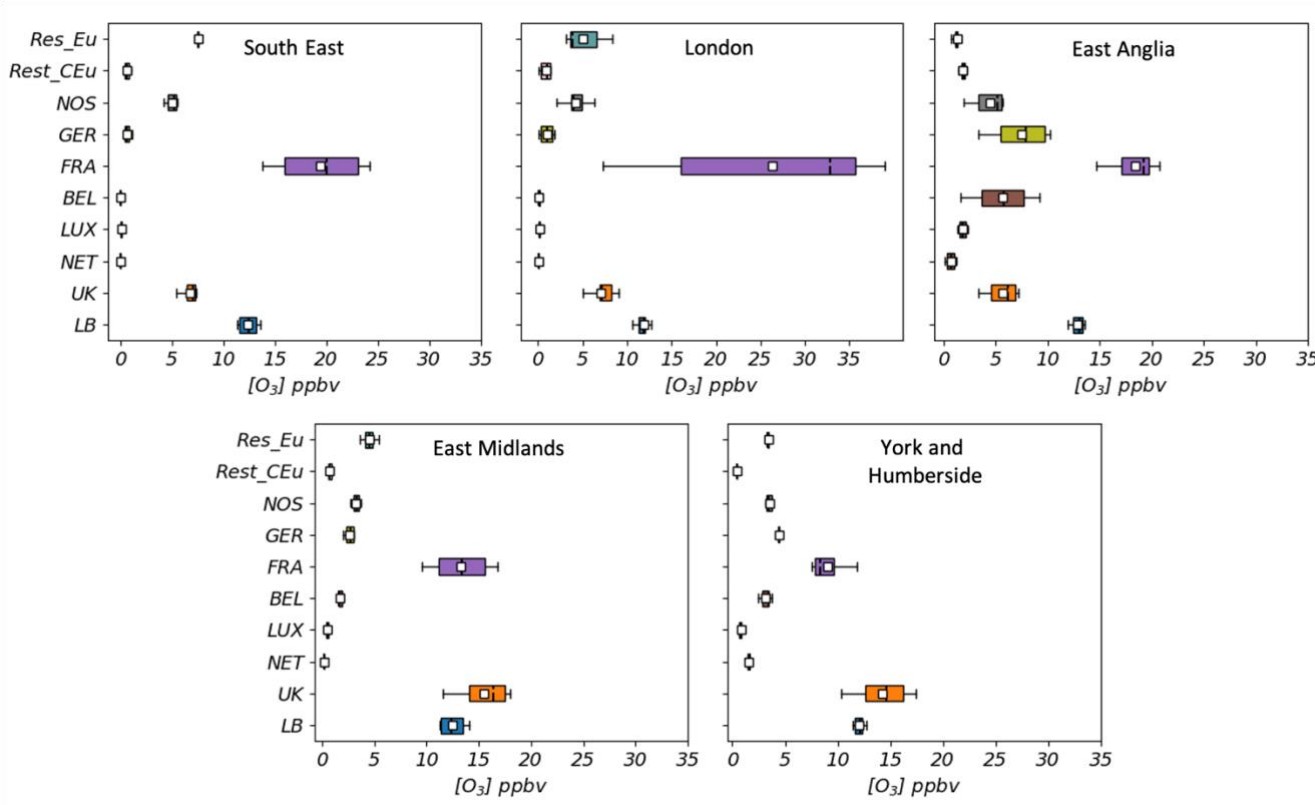


**Figure 11.** Hourly contributions, in ppbv, to surface O$_3$ at 5 UK receptor regions from 10 source regions (UK, background (LB), the Netherlands, Luxembourg, Belgium, France, Germany, rest of Central Europe (Rest_CEu), North Sea and English Channel (NOS) and rest of Eu) during days when the MDA8 is above 60 ppbv between May and August. The lower and upper ends of the boxes indicate the 25[th] and 75[th] percentiles, the black dashed line the median, the white boxes the mean and the whiskers the minima and maxima.





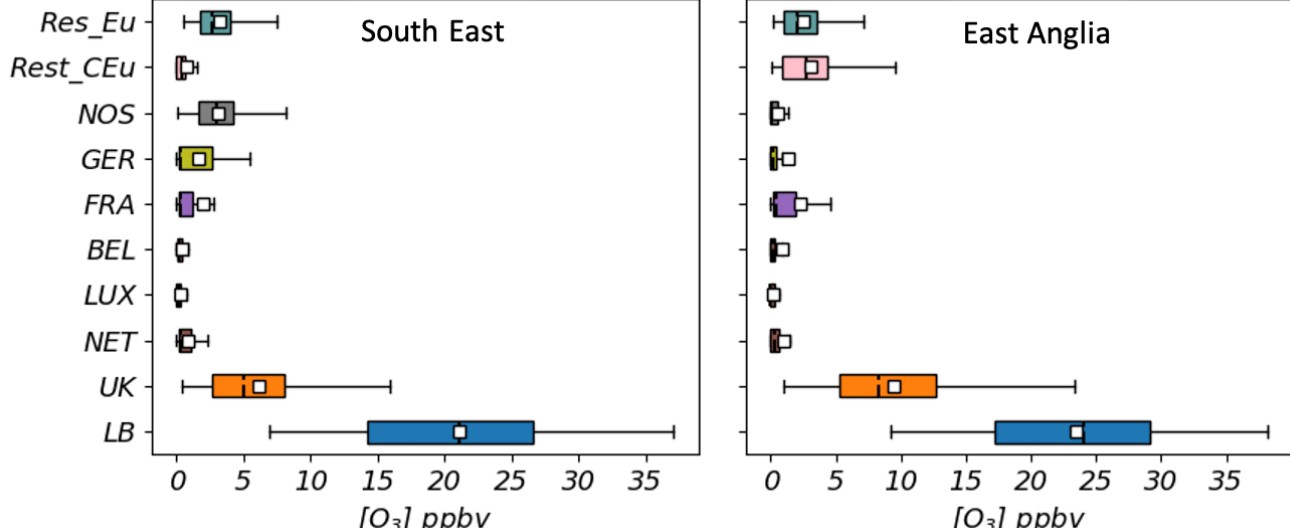

**Figure 12.** Hourly contributions, in ppbv, to surface O$_3$ at 2 UK receptor regions from 10 source regions (UK, background (LB), the Netherlands, Luxembourg, Belgium, France, Germany, rest of Central Europe, North Sea and English Channel (NOS) and rest of Eu) for AOT40 between May and August. The lower and upper ends of the boxes indicate the 25[th] and 75[th] percentiles the black dashed line the median, the white boxes the mean and the whiskers the minima and maxima.