# Peer review of "Sources of Surface O3 in the UK: Tagging O3 within WRF-Chem"

_Atmospheric Chemistry and Physics, 2022_

## Author Comment (AC1)

**RESPONSE TO REVIEWERS**

For 'Sources of Surface $O_3$ in the UK: Tagging $O_3$ within WRF-Chem' by Johana Romero-Alvarez, Aurelia Lupaşcu, Douglas Lowe, Alba Badia, Scott Archer-Nicholls, Steve R. Dorling, Claire E. Reeves, and Tim Butler

*This paper describes a modeling study to investigate the sources of ozone over the UK in the spring-summer period in 2015 using a tagged approach. It is a competent study using an established technique, and while the results are not unexpected, they provide a valuable quantification of source contributions that constitute one of the first available in the literature. In particular, the study highlights the importance of sources outside the region in influencing ozone, and provides a thorough quantification of local and regional contributions across different parts of the UK. The finding that different measures would need to be taken to address ozone as represented by the MDA8 and AOT40 metrics is interesting, and this finding could be exploited better in the paper. It also feels as though model evaluation has been skipped over lightly, and inclusion of a brief assessment to convince the reader of the quality of the model simulations would strengthen the paper. Once these issues have been addressed, along with the points below, I feel that the manuscript would make a valuable addition to the literature and would be suitable for publication in ACP.*

We appreciate the reviewer's positive assessment of the manuscript. Following the reviewer's recommendation, we have addressed all the comments to further strengthen the paper. Changes to the manuscript have been highlighted in yellow.

**General Comments**

*An original aspect of this study is consideration of impacts over different parts of the UK and using a number of different ozone metrics. Neither of these aspects is fully exploited in the results/discussion section, however. Which regions matter most from a population exposure perspective, for example? Which regions are currently close to regulatory limits? The exploration of different metrics is interesting, but how sensitive are the results likely to be to the meteorology in 2015? Sources in BEL/LUX/NET/GER may be more important than FRA in other years. Some consideration of these issues is needed.*

We have expanded the Results and Discussion section to include information about population exposure, regulatory limits, and dominant meteorology. The paragraphs below highlight the new information.

[revised manuscript text omitted]

*Evaluation of the model simulation is consigned to the supplement, but I feel that something is needed in the paper to convince the reader that the model is up to the task, particularly given that "a good representation of O3 in the European domain" is expressly stated in the conclusions. Please adapt the existing section 2.4 to provide a more quantitative summary of the model performance, particularly for O3 and NOx.*
*It would also be useful to show a 4-month timeseries of ozone at least one location to demonstrate the seasonal and diurnal variability (this could be hourly ozone or alternatively daily MDA8). This is important to show the relative importance of episodes, which are investigated in the latter part of the study.*

We agree with the reviewer's remark and adapted section 2.4 to include a quantitative summary of the model performance for $O_3$ and NOx, see below. We also added 4-months' time series for a costal site in East of England and two inland sites in south east of England.

'Table S.2 summarises the domain-wide statistical performance for NO, $NO_2$ and $O_3$. The predicted temporal correlation coefficient ($r$) for NO and $NO_2$ is fairly low (0.3), which is a feature exhibited also in other regional studies in Europe using WRF-Chem e.g., Tuccella et al. (2012), Pirovano et al. (2012) and Lupaşcu et a. (2022). The model underestimates NO mixing ratios in most analyzed sites with a domain-wide MB of -0.3 ppbv. $NO_2$ mixing ratios, on the other hand, are generally overestimated with a domain-wide MB of 0.31 ppbv, and no specific patterns distinguished in the bias distribution. This is consistent with the negative NO and positive $NO_2$ biases obtained across Europe using MOZART-4 chemistry reported in Mar et al. (2016).
The model's temporal variation in hourly $O_3$ concentrations at most sites is well represented, with an average $r$ value of 0.6. The model tends to underestimate concentrations in most locations, with a domain-wide mean bias of -3.7 µg m$^{-3}$. Correlation values above 0.5 are obtained in most sites, particularly in the UK, see Fig. S4a in the supplementary material. In contrast, low $r$ values (~0.4) are concentrated on high-altitude sites, which might indicate difficulties in the model representing $O_3$ transport. This is in line with previous studies using MOZART-4 chemistry, such as Knote et., (2014), showing low production of peroxyacetyl nitrates (PAN), an essential reservoir for $NO_2$ and a key player in remote $O_3$ production. Correlation values are consistent with summer time $O_3$ values below 0.40 reported on the WRF-Chem model evaluation over a European domain on Mar et al. (2016) using MOZART-4 chemistry.
Fig. 2 shows that the day-to-day variation in hourly $O_3$ mixing ratios is well represented by the model, except for large under-predictions during 1–3 July and 22-24 August, particularly at stations on the east coast, e.g., Weybourne. Note that the observed maximum hourly $O_3$ at this site is larger than those seen inland, e.g., Lillington Heath and Harwell (2015). This may indicate inflow of $O_3$ and precursors from nearby large metropolitan areas within the UK (e.g., London) or to longer-range transport from continental Europe. Thus, underestimation of $O_3$ during those days may be caused underestimation of long-range transport. This feature has also been identified in other source apportionment studies, such as Lupascu and Butler. (2019).'

*While the manuscript presents a case study from 2015, it would be valuable to speculate on how general the results are likely to be for other years.*

We agree with the reviewer's comment and added the following paragraph to the discussion:

'Notably, anticyclonic conditions in the UK have been associated with enhanced $O_3$ concentrations whereas cyclonic conditions and westerly winds have been linked to $O_3$ transport from the UK mainland and cleaner air from the North Atlantic (Jenkin et al., 2002; Pope et al., 2016; Romero-Alvarez et al., 2022). The contribution patterns described above may thus serve as predictors of future $O_3$ source apportionment over the UK regions.'

**Specific Comments**

*Line 40: narrow concentration window: this might be rephrased, as three orders of magnitude isn't particularly narrow.*

We agree with the reviewer. The paragraph has been rephrased as follow:

'The production of $O_3$ in the troposphere is highly non-linear. It depends on the abundance of nitrogen oxides (NOx = $NO_2$ + NO) and peroxy radicals ($HO_2$) generally produced after the oxidation of volatile organic compounds (VOCs) by hydroxyl radical (OH) (Monks, 2005). The reaction of NO with $HO_2$ and the subsequent photolysis of $NO_2$ generating $O_3$ is the primary known mechanism of $O_3$ production (Atkinson, 2000; Monks, 2005). NOx concentrations determine whether $O_3$ is produced or chemically removed (Monks, 2005). In the rural areas of most industrialized countries, where NOx is available at moderate levels, the rate of $O_3$ formation increases with increasing NOx concentrations (NOx-limited regime). In more polluted areas, by contrast, high NOx concentrations inhibit $O_3$ formation as this begins being depleted by NO (NOx titration effect). Subsequent formation of nitric acid ($HNO_3$) from the reaction of $NO_2$ with OH constitutes a major endpoint for $O_3$ in such environments (Monks, 2005). However, elevated inputs of non-methane VOCs (NMVOCs) can increase the production of $O_3$ as the reaction of VOCs with OH radicals become more significant (NOx-saturated regime).'

*Line 49: "European" -> "UK and European"*

The line was updated accordingly

*Line 56: Reductions in European NOx emissions would be expected to give a reduction in rural ozone concentrations in the UK, as this is far from the source region.*

Thank you for this observation, the paragraph has been reworded as:

'Accordingly, increasing emissions of precursors in Asia and North America influence $O_3$ concentrations entering Europe from the North Atlantic, offsetting the effects of European regional emission reductions on $O_3$ (HTAP, 2010; Derwent et al., 2018).'

*Line 65: As stated, tagged-ozone methods are better than perturbation approaches for attribution studies quantifying the contribution of different sources at a given place/time. However, they are less well suited for quantifying the effect*

*of emission controls which involve changing sources (which is how this concept was introduced in line 60). Some rephrasing is needed to avoid undermining the approach adopted here.*

The paragraph was updated as follows:

'S-R studies often compare model simulations that include all anthropogenic emissions with those obtained after modifying emissions from a region of interest (the so-called perturbation approach). However, as $O_3$ chemistry is highly non-linear, this approach can lead to unrealistic attribution estimates, e.g., Emmons et al. (2012) underestimated the $O_3$ contribution by up to a factor of 4 when perturbing NO emissions by 20%. Tagged-ozone methods, on the other hand, use additional diagnostics to follow the reaction of different emissions to the formation of $O_3$, making the approach suited to investigate the contribution of different precursors to the total amount of $O_3$ (Emmons et al., 2012; Grewe et al., 2012; Butler et al., 2018).'

*Line 75: Is the tagged ozone mechanism used here existing or new? Please make any novel aspects of the current study clear.*

The tagged ozone mechanism is the same used on Lupaşcu and Butler, (2019). However, in our setup we reduced the number of European sources that are tracked, and we do not explicitly track the HTAP regions that act as a boundary in Lupascu and Butler (2019). The line has been updated accordingly:

'The present study quantifies the contributions to surface $O_3$ in 12 receptor regions in the UK from anthropogenic NOx emissions from inside and outside the UK using the tagged-ozone method developed in Lupaşcu and Butler, (2019) with a reduced number of European source regions.'

*Line 107-8: It would be helpful to add a sentence here to suggest why nudging led to poorer simulations.*

We now give our interpretation of this effect in the paragraph, as follows:

'This decision was made after a test analysis showed that nudging of winds above the planetary boundary layer (PBL) and temperature at all layers, as done in Mar et al. (2016), leads to a representation of hourly $NO_2$ and $O_3$ mixing ratios in East Anglia region (East of UK) that was inconsistent with observations. The nudging simulation predicted shallower boundary layers compare with that obtained using the re-starting method, particularly over the Norfolk Sea coast, leading to high concentrations of $NO_2$, especially at night time, and larger $O_3$ lost due to increased deposition.'

*Line 126: "The method used here is based on...." Is the Lupascu and Butler approach used here directly or are there any developments or changes in implementation? It is important to be clear about the scientific contributions of the present study. Is any element of this new?*

The tagged ozone mechanism is the same used on Lupaşcu and Butler, (2019). However, in our setup we reduced the number of European sources that are tracked, and we do not explicitly track the HTAP regions that act as a boundary in Lupaşcu and Butler (2019). This is now clarified in the Methods section:

'The method used here attributes of $O_3$ contributions exclusively to NOx precursors using tagged-ozone method developed in Lupaşcu and Butler, (2019).'

*Line 136: How important is reentry of ozone into the model domain likely to be?*

The regional model is not combined with the global model as these are working as offline systems and therefore there is no feedback between the wrf and global model.

*Line 151: This sentence does not describe how the contribution of tagged O3 to AOT40 was calculated, it just describes how AOT40 is calculated.*

We agree with the reviewer and reworded the paragraph as follows:

'The AOT40 is defined as the accumulated excess of hourly $O_3$ concentrations above 40 ppbv measured during daylight hours (between 08:00 and 20:00) Central European Time (CET) over a typical three-month growing season May-July. Here, contribution of concentration of tagged $O_3$ to the cumulative metric AOT40 was calculated by selecting the hours when $O_3$ mixing ratios exceeded the hourly 40 ppbv threshold between 08:00 and 20:00 central European time (CET) from May-July over the most relevant arable farming areas in the UK, East Anglia and the South East, see Eq. (1).'

*Line 156: Equation 1 is incorrect: max(O3-40, 0)*

*Note that this is summed over specific hours, not all hours*

Equation has been corrected.

*Figures 7 and 8 show the same variable (O3 chemical production) and it would be helpful to combine them so that they can be compared more easily.*

Figures 7 and 8 have been combined as suggested.

*Figures 9-12: It is not clear that all four figures are required; presenting results for two contrasting months would be sufficient, with the others placed in the supplement. Note that use of contrasting color palettes would allow the reader to separate the inset pie chart more easily, and that separating the legend into two sections would make interpretation of the charts easier.*

Figures 9-12 had been updated as suggested. We also moved Figs. 10 to 12 to the supplementary material.

*Figures 13-15 could also be presented a lot more clearly, ideally with the panels arranged in a more geographically-intuitive layout. Flipping x and y axes would make the figures easier to read (so key sectors LB and UK are first rather than bottom of the list), truncating the O3 axis at 25 or 30 would make values more readable, and coloring bars consistent with Figs 9-12 would make contributions stand out better.*

Figures 13-15 had been updated.

**Typos and minor issues**

*Line 88: is -> are*

The line was modified accordingly.

*Line 94: citation error "G. a."*

The line was corrected 'Grell et al., 2005'.

*Line 100: citation format for Mar et al.*

The line was modified accordingly.

*Line 147: stablished (also exceeds -> exceed)*

The line was modified accordingly.

*Line 151: remove "concentration of"*

The line was modified as suggested.

*Line 169: units needed for the mean bias*

The line was changed accordingly ($\mu g \ m^{-3}$).

*Line 201: remove "from"*
The line was modified accordingly.

*Line 260: Remove subsection, as there is no 3.1.2*

The section was divided into two subsections: 3.1.1 Spatial distribution and temporal variation and, 3.1.2 Regional dependence.

*Line 321: Units on ozone mixing ratios*

The line has been corrected.

*Line 385: positive and negative bias in what/where?*

These refer to $O_3$ mean bias. This has been specified in the sentence.

*Line 506: The Romero-Alvarez reference is out of sequence*

The text has been corrected accordingly.

*The coastlines in Fig 1a are drawn at very low resolution, and the figure would look tidier if the resolution was improved. Consider adding the model grid to give the reader an indication of the model resolution.*

We have increased the resolution of the coast in the figure as suggested. However, we did not plot the model's grid cell to indicate the resolution as the tool we are using to create this plot do not support raster plots.

*Fig 6 caption: Closed up -> Close up*

The typo has been fixed.

*Data availability: key output data should be made available through a publicly accessible repository such as CEDA*

We will follow the Reviewer's suggestion and make the ascii files for the plots available on Zenodo.

*Author contributions: A clearer statement of author contributions in needed.*

We updated the authors contributions statement.

*Several entries in the reference list refer to discussion papers that are now published (e.g., Lupascu and Butler; Kuik et al.). Please update these.*

The references have been updated.

*Lines 798, 818: number not indicated in header, remove comment?*

We have removed the "number indicated in header" from Figures caption.

**Supplement:**

*S1.1: Person -> Pearson*

The typo has been fixed.

*p.5: particularly -> particularly*

The typo has been fixed.

*p.6: Fig 5S-> Fig S5, Fig 4S -> Fig S4*

The typos have been fixed.

*Most of the figures in the supplement are not of publication quality, and the timeseries in particular need to larger and more clearly labelled so that the comparison of measured and observed concentrations is clearer. In the spatial maps (Figs S6-S8) the results would be much clearer if a more appropriate color scale was used for the difference plots (ideally dichromatic).*

We have updated the figures as suggested.

*I do not find the composition comparison very convincing. While the analysis points to a number of model weaknesses, the causes remain unclear, so the comparison does not lend confidence in the performance of the model. While derived metrics, particularly those based on thresholds, are challenging to match well, I would have expected diurnal variation in NO, NO2 and O3 to be represented better.*

We understand the reviewer's concern. However, as pointed in Lupascu et al (2022) and the reference therein, several factors might be responsible for model performance, including relatively coarse resolution that increases diffusion into grid cells, and the errors associated with the wind speed and direction that can't capture reasonably well the transport of pollutant from the source. Moreover, a common feature of the models is the overestimation of nighttime NOx concentration (Kuik et al., 2018, Im et al., 2015) due to reduced mixing at nighttime.

Lupaşcu, A., Otero, N., Minkos, A., and Butler, T.: Attribution of surface ozone to NOx and VOC sources during two different high ozone events, Atmos. Chem. Phys. Discuss. [preprint], https://doi.org/10.5194/acp-2022-189, in review, 2022.
 Kuik, F., Kerschbaumer, A., Lauer, A., Lupascu, A., von Schneidemesser, E., and Butler, T. M.: Top–down quantification of NOx emissions from traffic in an urban area using a high-resolution regional atmospheric chemistry model, Atmos. Chem. Phys., 18, 8203–8225, https://doi.org/10.5194/acp-18-8203-2018, 2018.
Im, U., Bianconi, R., Solazzo, E., Kioutsioukis, I., Badia, A., Balzarini, A., Baró, R., Bellasio, R., Brunner, D., Chemel, C., Curci, G., Flemming, J., Forkel, R., Giordano, L., Jiménez-Guerrero, P., Hirtl, M., Hodzic, A., Honzak, L., Jorba, O., Knote, C., Kuenen, J. J., Makar, P. A., Manders-Groot, A., Neal, L., Pérez, J. L., Pirovano, G., Pouliot, G., Jose, R. S., Savage, N., Schroder, W., Sokhi, R. S., Syrakov, D., Torian, A., Tuccella, P., Werhahn, J., Wolke, R., Yahya, K., Zabkar, R., Zhang, Y., Zhang, J., Hogrefe, C., and Galmarini, S.: Evaluation of operational on-line-coupled regional air quality models over Europe and North America in the context of AQMEII phase 2. Part I: Ozone, Atmos. Environ., 115, 404–420, https://doi.org/10.1016/j.atmosenv.2014.09.042, 2015.

---

## Author Comment (AC2)

**RESPONSE TO REVIEWER No 2**

For 'Sources of Surface $O_3$ in the UK: Tagging $O_3$ within WRF-Chem' by Johana Romero-Alvarez, Aurelia Lupaşcu, Douglas Lowe, Alba Badia, Scott Acher-Nicholls, Steve R. Dorling, Claire E. Reeves, and Tim Butler

*This paper describes the application of a regional chemical transport model using an ozone tagging scheme to quantify source contributions to surface tropospheric ozone in the UK during May-Aug 2015. The application of such a scheme in this context is novel, and the paper provides useful insight into the local, wider European, and extra-European contributions to ozone, broken down by local region within the UK. The paper explores differences in source contributions during episodes of higher surface ozone concentrations, and explores contributions using air quality and vegetation-relevant metrics, which provide some policy-relevant context. The paper is well written, and the methods applied appear robust and well described. There are some aspects of the model information and evaluation, and well as improvements in the discussion of results that would improve the manuscript. I recommend that once these issues (described below) are addressed, that the paper be published in ACP where it will be a valuable addition to the literature on European ozone air quality.*

We appreciate the reviewer's positive assessment of the manuscript. Per the suggestion, we have carefully edited the document to improve presentation of results and discussion. Changes to the manuscript have been highlighted in yellow.

**General comments**

*For high ozone episodes in summer, biogenic emissions may be an important driver of ozone formation (e.g. see point made in Introduction on Page 7). Even if it is not possible to evaluate the model-simulated isoprene with observations, it might be informative to include a supplementary plot of isoprene during high ozone and more average conditions. The authors could also refer to previous studies evaluating MEGAN isoprene emissions in WRF-Chem, if relevant.*

We thank the reviewer for this recommendation. In the supplementary material, we have included a time series (Figure S.9) comparing measured and modeled isoprene during July 2015 in the East Anglia region. We also added the following paragraph:

'The model's representation of organic NMVOCs may be an additional source of bias. Figure S.9 in the supplementary material shows that the model largely underestimates observations of isoprene particularly during the first days of July which were characterized by high $O_3$ mixing rations. The impacts of isoprene chemistry in $O_3$ concentrations have been reported largely in the literature. For instance, in box modelling studies, Knote et al. (2014) show large variations in isoprene concentrations between different chemical mechanisms despite using identical biogenic emissions. Moreover, Zhao et al. (2016) demonstrate that more recent versions of the Model of Emissions of Gases and Aerosols from Nature (MEGAN) better reproduce the observed isoprene than the publicly available version of the MEGAN model integrated into WRF-Chem.'

*Is it possible to calculate population-weighted MDA8 ozone contributions using population data and the model output? This would really strengthen the relevance of the results to air quality and human health. At the moment the discussion does not differentiate based on population distributions among the different regions, so it is difficult to interpret the relevance of the results to air quality.*

We appreciate the reviewer's suggestion. Although assessing the population exposure to MDA8 $O_3$ is beyond the scope of this paper, we have reinforced the discussion based on the regions that matter most from a population exposure perspective, see below.

'The mean contribution from the Eu super-region (FRA, GER, NET, LUX, BEL, NOS, Rest_CEu and Rest_Eu) accounts for nearly 16 % of the simulated monthly mean $O_3$. The largest Eu super-region contributions are observed in the UK locations closer to the continental Europe and that together contain about 40% of UK population (East Anglia, London area, South-East England and Yorkshire)'

'The LB is the principal contributor to the modelled mean $O_3$ mixing ratios in every receptor region. The contributions peak in May (mean absolute contribution 25 ppbv), reflecting the seasonal cycling in the northern hemispheric background $O_3$ (e.g., Monks, 2000; AQEG, 2009). Contributions from this source are more prominent in the regions located in the north, east, and north-west of the UK, e.g., Scotland (30 ppbv), Northern Ireland (28 ppbv), North-East (27 ppbv), the North-West, and Wales (26 ppbv). These regions contain about 20% of UK population and are primarily impacted by westerly flows and associated hemispheric $O_3$ background due to their geographical positions (AQEG, 2009). Also, they generally experience less than 10 days with $O_3$ concentrations above the EU limit of 120 μg m$^{-3}$ (DEFRA 2020) because of low NOx emissions locally'

'The UK contributions are generally more significant in the east, south-east, and the Midlands, showing a maximum value in June and July in every receptor area, figures S.9 and S.10 in the Supplemental Material. The source region provides up to 20% of the surface $O_3$ mixing ratios in East Anglia, 18% in the London area and East Midlands, and 16% in Yorkshire and the South East, making it the second-biggest source of $O_3$ in these locations after the LB. This area incorporates about 50% of UK population and often experiences more than 10 days with $O_3$ concentrations above the EU and UK threshold (concentration > 120 and 100 μg m$^{-3}$) (DEFRA 2020).'

'The mean contribution from each source region for the hours when the MDA8 $O_3$ exceeded 50 ppbv at each receptor area from May to August is presented in Fig. 10. The figure shows large contributions from source regions that were not seen as dominant sources. France, for example, becomes a major source, particularly in receptors in densely populated areas such as the south and east of the UK.'

*During ozone episodes (presented as when MDA8 O3 exceeds 50 or 60 ppbv), it would be informative to provide more in-depth discussion of meteorological conditions alongside the source region contributions. Are these periods dominated by anticyclonic conditions? What are the atmospheric transport pathways that dominate the France-sourced O3 influence on UK ozone? Are there any specific features that characterize the MDA8 > 60 ppbv episodes from the more moderate 50 ppbv exceedances?*

We thank the reviewer for this observation. We have added information regarding the predicted meteorological conditions during the $O_3$ episodes (MDA8 above 50 and 60 ppbv) as suggested. Below are the paragraphs that have been modified in Results and Discussion.

'The summer months see an increase in the input from France, Germany and the Benelux region, in particular during anticyclonic weather conditions and over the receptor regions located in the south and east of the UK (e.g., South East England, East Anglia, the London area and the East Midlands). This is consistent with results of studies on extreme $O_3$ in the EU and the UK reporting an increase in surface $O_3$ concentrations under anticyclonic conditions (e.g., Pope et al. (2016); Ordóñez et al. (2016); Romero-Alvarez et al. (2022)). Romero-Alvarez et al. (2022), in particular, has

shown that a wide area of high pressure centred over the Netherlands coast affected most of England during the first days of July 2015. During the same period, regions such as the East Anglia reported increases in $O_3$ mixing ratios of up to 16.6 ppbv h$^{-1}$ that overlapped with wind direction changes from south-southwest to south-southeast. Depending on the predominance of the wind direction (south- southeast and south-southwest), $O_3$ from anthropogenic sources within France can impact both the west and the east of the UK.'

'The contribution is greater in the southern UK due to the proximity to the source region. The contributions from the Benelux region and Germany are more significant in the east of the UK due to the proximity with the continent and association with easterly flows (east and southeast) (about 14% and 6% of the Eu super-region in the East Anglia during the summer months comes from these two source regions, respectively).'

'France was the most significant contributor to $O_3$ build-up when the mixing ratios exceeded the EU threshold in South East England (mean ~18 ppbv), East Anglia (mean ~21 ppbv), and the London area (mean ~26 ppbv) because convergence of westerly and south-easterly winds in the west of the UK diverted the contributions of domestic sources from these regions, as reported in Romero et al., (2022).'

'As in the contributions to the MDA8 $O_3$ value of 50 ppbv above, the lateral boundary component remained nearly constant in all receptor areas with a mean contribution of about 12 ppbv. This is because most of the UK's weather was dominated by anticyclonic conditions.'

'When exceedances to the hourly surface $O_3$ mixing ratios above 40 ppbv is considered, the LB component becomes the dominant source in both receptor regions (estimated mean concentration between 21-24 ppbv) as its threshold is close to the tropospheric baseline ozone level associated with maritime Nort Atlantic air masses.'

'Romero-Alvarez et al. (2022) has shown that MDA8 $O_3$ above 50 ppbv in the Southeast and East Anglia regions coincided in July 2015 with days when easterly winds prevailed (east-southeast flows). In contrast, MDA8 $O_3$ above 60 ppbv coincided with a shift in the wind direction from east-southeast to south-southeast and south and a sharp rise in the surface temperature.'

**Specific Comments**

*Introduction - be more explicit about describing ozone production dependencies in NOx and VOC-limited conditions, and importance of NO+O3 in high NOx environment. This effect is variously referred to as 'titration' and 'scavenging'. It would help the reader to point out the reaction specifically.*

The introduction now includes the $O_3$ titration reaction and an extended description of the NOx and VOC-limited ozone formation regimes. We also added $HNO_3$ formation to highlight ozone titration as a loss mechanism for ozone:

'The production of $O_3$ in the troposphere is highly non-linear. It depends on the abundance of nitrogen oxides (NOx = $NO_2$ + NO) and peroxy radicals ($HO_2$) generally produced after the oxidation of volatile organic compounds (VOCs) by hydroxyl radical (OH) (Monks, 2005). The reaction of NO with $HO_2$ and the subsequent photolysis of $NO_2$

generating $O_3$ is the primary known mechanism of $O_3$ production (Atkinson, 2000; Monks, 2005). NOx concentrations determine whether $O_3$ is produced or chemically removed (Monks, 2005). In the rural areas of most industrialized countries, where NOx is available at moderate levels, the rate of $O_3$ formation increases with increasing NOx concentrations (NOx-limited regime). In more polluted areas, by contrast, high NOx concentrations inhibit $O_3$ formation as this begins being depleted by NO (NOx titration effect). Subsequent formation of nitric acid ($HNO_3$) from the reaction of $NO_2$ with OH constitutes a major endpoint for $O_3$ in such environments (Monks, 2005). However, elevated inputs of non-methane VOCs (NMVOCs) can increase the production of $O_3$ as the reaction of VOCs with OH radicals become more significant (NOx-saturated regime).'

*Line 79: Not clear what is meant by "the second warmest year in a row in Europe".*

The sentence has been removed as it was not longer relevant.

*Line 80: "EU information threshold of 1 hour (h) average mixing ratio of 180 µg m-3": the value of 180 µg m-3 is a concentration not a mixing ratio. Is the threshold defined as the 90 ppbv mixing ratio, or the 180 µg m-3 concentration? These are not necessarily equivalent (dependent on local meteorological conditions).*

Thank you for the observation. The EU information threshold is defined as 90 ppbv (mixing ratio). We modified the sentence as follows:

'Several heat waves causing elevated $O_3$ values in Central and Western Europe that exceeded the EU information threshold of 1 hour (h) average mixing ratio of 90 ppbv.'

*Line 97-99: Please clarify how the IC concentrations are applied. The phrase implies that they are used to initialise the model simulation at the outset, however the text implies that they are applied every 3 hours. Does this mean that the model fields are essentially overwritten with MOZART fields every 3 hours? Please clarify.*

With thank the reviewer for the observation. Only BCs are ingested by the model every 3 hours. We have changed the sentence as follows:

'IC and BC for the chemistry fields were extracted from global simulations produced by the Chemistry Transport Model for $O_3$ and Related Chemical Tracers MOZART-4 GEOS-5 (Emmons et al., 2010). BCs were ingested into the model every 3 hours.'

*Line 103: Presumably aerosol are also simulated in the model? Please provide information on the aerosol scheme used in the simulations.*

Simulations were conducted only for gas-phase chemistry.

*Line 169: Mean bias in µg m-3, ppb, or %? Please clarify.*

Mean bias is in $\mu g\ m^{-3}$. We changed the line as follows: 'domain mean bias (MB)= -3.71 $\mu g\ m^{-3}$'

*Figure S1 - Do you have an explanation for the lack of diurnal cycle in the model surface temperature at coastal sites? Does this imply issues regarding diurnal variation in mixing height / boundary layer? Is there any potential link to biases in the NOx and ozone shown? It would be helpful to expand more on some of these evaluations and comparisons in the main text.*

The temperature at coastal stations is strongly influenced by sea surface temperature, therefore the coastal air temperatures are less variable than inland temperatures. We checked the land cover map and the selected grid cells and they correspond to a water body (ocean). We therefore removed the panels from the plots and corresponding statistical analysis as it is no longer relevant analysis.

*Fig. 3, 4, 6 captions: the plots depict mixing ratio, not concentration. Please change wording to reflect this.*

Caption in Figs. 3, 4 and 6 was changed to mixing ratio.

**Typographical errors:**

*Line 35: "Concentration of …" -> "The concentration of.."*

The line was corrected as suggested

*Line 94: Erroneous "G. a."?*

The line was corrected 'Grell et al., 2005'

*Line 100: "shipping lines' -> "shipping lanes"?*

The line was corrected as suggested

---

## Author Response (AR2)

**RESPONSE TO EDITOR**

For 'Sources of Surface $O_3$ in the UK: Tagging $O_3$ within WRF-Chem' by Johana Romero-Alvarez, Aurelia Lupaşcu, Douglas Lowe, Alba Badia, Scott Archer-Nicholls, Steve R. Dorling, Claire E. Reeves, and Tim Butler

*Comments to the author:*
*Dear Authors,*
*Thank you for submission of a revised manuscript and your response to the referees comments. I think the paper is basically ready for publication, but a technical issue remains. The data availability statement in the manuscript mentions modifications made to the WRF model that are "introduced and described in Section 2". It is not clear to me which modifications you mean. The modifications should be clearly explained in Section 2. Furthermore, in the data policy statement you mention "The modification introduced and described in Section 2 as well as the model data can be provided upon request." According to the data policy of ACP, such information and data should be made available, for example, in the Supplement or in a public repository (https://www.atmospheric-chemistry-and-physics.net/policies/data_policy.html). Please provide the requested information, accordingly.*

*Non-public comments to the Author:*
*Please proceed as discussed in our emails.*

We appreciate the editors' positive assessment of the manuscript.
A paragraph describing the model's modifications needed to accommodate the ozone tagging mechanism in WRF-Chem has been included in session 2.
The chemical mechanism, as well as the model output, are now available online via Zenodo at https://doi.org/10.5281/zenodo.6968040 and https://doi.org/10.5281/zenodo.6968649